



# Modulation of Polar Mesospheric Summer Echoes (PMSE) during HF Heating

Tinna L. Gunnarsdottir [1], Arne Poggenpohl [1,2], Ingrid Mann [1], Alireza Mahmoudian [3], Peter Dalin [4], Ingemar Haeggstroem [5], and Michael Rietveld [6]

[1]Department of Physics and Technology, UiT Arctic University of Norway, Tromsø, Norway
[2]Physics Faculty, TU Dortmund University, Germany
[3]Institude of Geophysics, University of Tehran, Iran
[4]Swedish Institute of Space Physics, IRF, Kiruna, Sweden
[5]EISCAT Scientific Association, Kiruna, Sweden
[6]EISCAT Scientific Association, Ramfjord, Norway

**Correspondence:** Tinna Gunnarsdottir (tinna.gunnarsdottir@uit.no)

**Abstract.**

The formation of Polar Mesospheric Summer Echoes (PMSE) is linked to the presence of charged dust/ice particles in the mesosphere and the PMSE modulation is a possible way to observe the effect of dust charging. We investigate the modulation of PMSE by HF radio waves based on measurements carried out with the EISCAT Heating facility and the EISCAT VHF radar in

August 2018 and August 2020 toward the end of the PMSE season. The measurements were made during the night with reduced solar illumination. The EISCAT Heating was operated in subsequent 48 s on and 168 s off intervals. In our observations, the PMSE modulation by the HF heating disappears during ionospheric conditions of energetic particle precipitation. We observe more than half of the cycles being influenced by the heating with a reduced PMSE power when the heater is on and a similar amount of cycles showing an increase in power when the heater is turned off. In less than half of the observed cycles we see an

overshoot and it seems to be largely influenced by the overall PMSE power; with smaller or nonexistent overshoot when the PMSE power is high. We observe instances of very large overshoots, where background PMSE power seems to be reduced. During periods when we observe modulation, they often vary strongly from one cycle to the next; they are highly variable on spatial scales smaller km and time scales of minutes that are shorter than the scales assumed for the variation of dust parameters. Averaged curves over several heating cycles are similar to the overshoot curves predicted by theory and observed previously.

Some individual curves however, show a stronger overshoot than observed in previous studies. A possible explanation for this difference can lie in the dust charging conditions that are different during the night or other conditions might be at play. We observe two possible instances of sporadic E-layers that are influenced by heating but do not show overshoots, as is to be expected.





**Contents**



# 1 Introduction

Polar mesospheric summer echoes (PMSE) are strong coherent radar echoes from 80 to 90 km altitude that are observed at
high and mid latitude during summer. It was first noted in the 1970s that the coherent radar echoes that were observed during
the year were unusually strong and from unusually high altitude during the summer (Ecklund and Balsley, 1981; Czechowsky
et al., 1979); and that they originate from the height of the extreme temperature minimum around the mesopause that occurs
at high and mid-latitudes in summer (Ecklund and Balsley, 1981). Later, the echoes were observed from other places and with
radars with frequencies ranging from 50 MHz – 1.3 GHz (Cho and Röttger, 1997). The PMSE are observed from mid May
to end of August in the northern hemisphere with main occurrence during local noon (Latteck et al., 2021) coinciding with
maximum solar illumination. The observed reflection of the radio waves results from strong variations in the electron density
and thus the refractive index. The echoes are particularly strong because the backscattered radio waves interfere constructively
when the distance between the scattering centers is half the wavelength, which is called the Bragg condition. Scattering at the
Bragg condition is typically caused by neutral turbulence in the atmosphere. In case of the PMSE it arises from a combination of
neutral turbulence and the presence of charged ice particles that form near the mesopause and influence the electron distribution;
the presence of the ice particles expands the Bragg scales for which the echoes are observed (Rapp and Lübken, 2004). The
spatial distribution of the ice particles at these altitude is influenced by the complex neutral atmosphere dynamics caused by
the upward propagating gravity waves and can also be seen in the structure of the noctilucent clouds(NLC) (Dalin et al., 2004).

The region of PMSE occurrence overlaps with that of noctilucent clouds, which are an optical manifestation of ice particles.
Temperature studies of the summer Arctic mesosphere suggest that both phenomena are temperature controlled and occur at
temperatures $130 - 150$ K and lower around the mesopause (Lübken, 1999) where water ice particles can form. Since 2007, the
water ice particles have also been observed by satellites in so-called polar mesospheric clouds (PMCs); and the measured cloud
extinctions are explained with the optical properties of water ice with inclusions of smaller meteoric smoke particles (Hervig
et al., 2012). The meteoric smoke particles are nanometer sized dust particles that form from ablated meteoric material in the
altitude range $70 - 110$ km (Rosinski and Snow, 1961; Hunten et al., 1980; Megner et al., 2006). The satellite observations
also supported the already existing hypothesis that the ice particles are formed by heterogeneous condensation, which has just
been supported by a study that applies a new theoretical condensation model (Tanaka et al., 2021). The surface charging of
dust particles, be it meteorite smoke or ice particles or a mixture of both, is an important process that influences the growth of
particles and at the same time gives clues to their size and composition (Rapp and Thomas, 2006). The charges and sizes of the
dust particles are largely unknown.

Many authors have suggested that the modulation of PMSE during artificial heating with HF radio waves could be used to
study the underlying plasma and dust particles (Biebricher et al., 2006; Mahmoudian et al., 2011, 2020). During such heating
experiments the electron temperature is locally and temporarily enhanced (Rietveld et al., 1993); and it was first noticed by
Chilson et al. (2000) that PMSE can be modulated during such heating. The PMSE often almost disappear when the heater
is turned on and then return when the heater is turned off again. It is assumed that the increased electron temperature during
heating and the resulting increased diffusion reduces the fluctuations in the electron density and thus the PMSE power (Rapp





and Lübken, 2000). Havnes (2004) found that with an adequate on/off time of the heater, a so called overshoot characteristic curve could be generated, in which the PMSE power did not return to the original value after heating, but exceeded it. Such overshoot curves have been observed in many simultaneous radar and Heating studies of PMSE made with EISCAT. The

overshoot curves have also been observed for some polar winter mesospheric echoes (PMWE) (Kavanagh et al., 2006; Belova et al., 2008; Havnes et al., 2011). Most PMWE do not appear to be associated with the presence of dust (Latteck et al., 2021), but those showing overshoot are more likely associated with the presence of small dust, possibly meteoric smoke. Therefore, studies of PMWE modulation during heating could be considered as informative as those of PMSE.

With this work we want to investigate whether and how the PMSE modulation during heating can be used for systematic

investigations of the charged dust component. We present observational studies of PMSE with the EISCAT VHF radar during four VHF/Heating campaigns which were all done in August during twilight or night conditions. And we investigate the modulation of the PMSE during HF heating.To our knowledge this is the first systematic investigation of PMSE modulation under reduced photo-emission conditions during the night and toward the end of the PMSE season. The remaining manuscript is structured as follows: First, the section 2 introduces the PMSE modulation during heating and the overshoot effect. Section 3

describes the experiments we performed including the radar and heating parameters and gives an overview of the observational results. Then a discussion of the PMSE modulation is given in section 4 where we first give an overview of the observed PMSE modulation and then discuss in particular the cases of quiet ionospheric conditions and of an ionosphere that is moderately influenced by energetic particle precipitation. We make a comparison with a model calculation and discuss the overall outcome. A short conclusion is given 2 and additional information on observational data is given in the appendix.

## 2   PMSE and Heating

The EISCAT Heating facility transmits high-frequency radio waves of high power into the atmosphere (Rietveld et al., 1993). Electron oscillations that are associated with the wave absorption translate into thermal motion, heating the electron component while the other plasma components keep their initial temperature. As was mentioned above it was found that this active heating influences the PMSE signal. During the experiments the heating is in a sequence switched on and off in pre-defined time

intervals (here we use 48 seconds on and 168 seconds off) and the PMSE echoes are simultaneously observed with the EISCAT VHF radar. The time variation of the observed PMSE power is sketched in Fig. 1 to illustrate the observed phases of the PMSE heating cycle and the often seen overshoot curve: decline, heating phase, recovery/overshoot and relaxation.





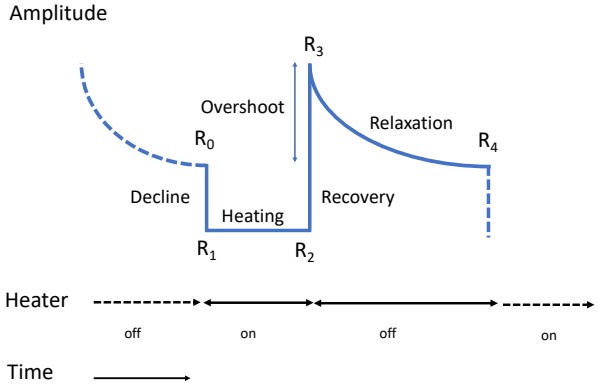

**Figure 1.** Sketch of the PMSE modulation due to HF heating in a typical overshoot curve; the power amplitudes during different times of the heating cycle are defined.

**Decline** - $R_0 \rightarrow R_1$: As the heater is switched on at $R_0$ the power effectively falls off instantaneously (depending on the radar frequency used) (Havnes, 2004). The back-scattered power drops because the heating enhances the electron temperature

and consequently the electron diffusivity so that the large electron density gradients are reduced and therefore the backscatter is less efficient (Rapp and Lübken, 2000).

**Heating** - $R_1 \rightarrow R_2$: During the heater on phase from $R_1$ and $R_2$ there are some variations in the power amplitude. Because of the higher electron temperature the charging electron flux on the dust particles increases during the heater on period and often an increase in the power can be seen. The charging timescales become shorter and compete more with the faster electron

diffusion (Mahmoudian et al., 2011).

**Recovery/overshoot** - $R_2 \rightarrow R_3$: The power then increases when the heater is switched off (recovery) and in many cases the power rises above the previous undisturbed level (overshoot). When the heater is switched off again, the electron temperature drops quickly to the initial value before the heater was on due to the highly collisional regime present at these altitudes. The dust particles carry a higher charge than before and therefore repel the electrons more strongly. The electrons follow the ion

diffusivity and as a result, the electron density gradients are larger than initially and therefore the backscatter is higher, creating an overshoot in the power.

**Relaxation** - $R_3 \rightarrow R_4$: During the next phase the power relaxes back to the previous undisturbed level. With a varying relaxation time depending on the conditions. With a long relaxation time, new and undisturbed plasma can enter the radar beam or the dust present has time to discharge (Havnes, 2004) .

The amplitudes ($R_0$, $R_1$, $R_2$, $R_3$ and $R_4$) marked in Fig. 1 will be considered in our analysis of the observations below, where $R_4$ is then the start ($R_0$) of the next subsequent cycle. We follow previous studies and refer to the curves that describe the measured PMSE during one heating cycle (On and Off time) as overshoot curves.



## 3   Observations

We first describe the overall observation conditions, radar operation and radar analysis and present an overview of the data.

### 3.1   Overall observation conditions

The presented observations were carried out during the "Mesoclouds 2018" and "Mesoclouds 2020" campaigns conducted in a collaboration between UiT Tromsø and IRF Kiruna. The EISCAT VHF radar and the EISCAT Heating facility are located in Ramfjord near Tromsø, Norway (69.59°N, 19.23°E). The observations were made on 11/12 August 2018, 15/16 August 2018, 05/06 August 2020, 06/07 August 2020, respectively during the night between 20:00 and 02:00 UT. Most observations of PMSE and heating have been done around noon and often during high summer, June and July. These observations thus represent dusk and night conditions with reduced influence of sunlight on the observational volume compared to other observations. We estimated the relative difference an observation done on 21 of June (summer solstice) at noon receives of sunlight compared to our observations for 11 and 15 of August (2018) at 21h and 22h (UT) (see e.g Giono et al. (2018) for how this can be done). Where we used solar flux for the Lyman-$\alpha$ line (121.56 nm) from the SOLSTICE instrument on the SORCE satellite (https://lasp.colorado.edu/home/sorce/data/ssi-data/) and neutral density for $O_2$ from the NRLMSISE-00 Atmosphere Model (Hedin, 1991) for the location of the EISCAT VHF radar. We estimate that the solar illumination is reduced by at least one order of magnitude in comparison to the observation conditions in June during the day. This influence should translate to a reduced photo-emission current and thus influence the dust charging conditions.

Simultaneous optical measurements of NLC were done using two optical NLC cameras located at Kiruna and Nikkaluokta (Sweden). Both locations of the NLC cameras are about 200 km south of Tromsø, which permits making NLC triangulation measurements. There was however no NLC observation above the radar site, mainly because of weather conditions. There was bad weather on 11/12 August (clouds) both in Kiruna and Nikkaluokta. During the night of 15/16 August, faint NLC were observed from Kiruna close to the horizon, approximately above Andøya, i.e. more westward than the EISCAT site. Weather conditions prohibited NLC observations during the 2020 radar campaigns.

Fig. 2 a and b show the temperature profiles (blue line) as measured by the Aura satellite and frost point temperature profiles (green line) estimated using the Aura water vapor data (both the temperature and water vapor were measured with the Microwave Limb Sounder, MLS instrument). These profiles have been selected to be closest ones to the radar location and to the times of the PMSE observations. The height ranges in which the temperature is lower than the frost point temperature indicate the regions where ice particles can form. This gives good indication of the conditions present in the atmosphere, showing that the temperatures are cold enough to facilitate ice particle formation at PMSE altitudes, however there could be variations due to the spatial and temporal difference between the measurements that must be kept in mind.



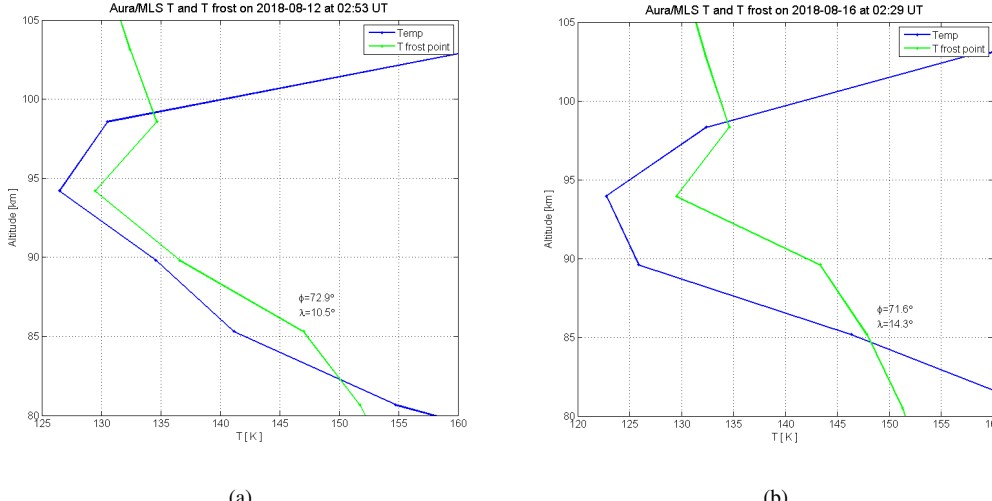

(a)                                          (b)

**Figure 2.** Temperature profiles (blue line) as measured by the Aura satellite at 12.08.18 and 16.08.18, and frost point temperature profiles (green line) estimated using the Aura water vapor data. Closest measurements in space and time.

### 3.2 Radar operation and data analysis

The radar observations were made in zenith direction with the EISCAT VHF (224 MHz) antennas near Tromsø (69.5864°N, 19.2272°E). The radar code used was Manda and reference to EISCAT documentation and radar and heating system parameters

are given in Table 1. The EISCAT Heating (Rietveld et al., 1993, 2016) was operated with a vertical beam at 5.423 MHz with a nominal 80 kW per transmitter which corresponds to Effective Radiated Power (ERP) in the range between 500 and 580 MW and X-mode polarization were used with a sequence of 48 s on and 168 s off. The vertical extension of the heater beam extends far beyond the region covered by the radar. Given that the vertical winds and velocity fluctuations of the PMSE observed with EISCAT VHF are within few m/s and horizontal winds possibly few 10 m/s (Strelnikova and Rapp, 2011), the radar at all times

measures PMSE that are influenced by the heating.

A standard incoherent scatter analysis, GUISDAP (Lehtinen and Huuskonen, 1996) was used to derive the radar data products. It provides the electron density derived from the incoherent scatter spectrum and a value proportional to back-scattered power for the coherent scatter of the PMSE. We therefore use the unit of equivalent electron density as was done previously for observations of PMWE (Kavanagh et al., 2006; Belova et al., 2008) and PMSE (Mann et al., 2016). The post-experiment

integration time used throughout this analysis was 24 s for computational reasons except for one of the observations when we compare with simulations, then a resolution of 4.8 seconds was used. We found that choosing a higher time resolution for the overall discussion did not result in additional information. We consider the back-scattered power in units of equivalent electron density throughout this paper.





**Table 1.** Parameters for EISCAT VHF radar operation and EISCAT heating facility. Half of the VHF radar is used for transmitting and entire radar used for receiving(beam width adjusted accordingly)

| EISCAT VHF | | | |
|---|---|---|---|
| Frequency | 224.4 MHz | Resolution in range | 360 m |
| Wavelength | 1.34 m | System Temperature | 240 – 370 K |
| Transmitter peak power | 1.5 MW | Antenna gain | 43 dBi |
| Radar Code | Manda | Half-power beam width | 1(2) x 2.4 x 1.7 ° |
| **EISCAT Heating Facility** | | | |
| Frequency | 5.423 MHz | ON time | 48 s |
| Beam Width | 7 ° | OFF time | 168 s |

### 3.3 Overview of observations

The observations were made from 20:00 to 02:00 UT during four nights in August 2018 and 2020. The observations are displayed in Fig. 3 shown for the entire time period and with altitudes from $80-110$ km. This altitude range allows us to show the PMSE present as well as other relevant information, like particle precipitation and possible sporadic E-layers. White vertical areas are observation gaps due to operation problems. In each data set we identified interesting measurement intervals that we considered for analysis. A closer look at each area is given in Appendix A as well as an overview of the time and 175 altitude range of the areas are given in Table A1.

**Observation 1: 11/12 August 2018** During the first night PMSE were observed until around 01:30 am. One can see that the electron densities above and partly below the PMSE are high, showing typical appearance of particle precipitation. In area 1 the precipitation is especially strong and enhanced electron density observed as low as 80 km, well below the PMSE layer. We considered:

– Area 1: PMSE with strong precipitation in the altitude range $83.4-85.6$ km from 21:36 UT lasting about 20 heater cycles

   – Area 2: high altitude and long lived PMSE layer extending from $86.3-90$ km during about 40 heater cycles starting from 23:06 UT with some precipitation.

   – Area 3: low altitude PMSE at $83.4-86.4$ km from 00:00 UT lasting about 30 heater cycles with some precipitation at end of layer.





**Figure 3.** Overview of all four observation days with time intervals and dates given in each respective figure.





**Observation 2: 15/16 August 2018.** During the second observation, PMSE were observed before midnight and then again at 2:00 am at the end of the measurements. A first observed PMSE (Area 1) seems not influenced by precipitation, it almost seems that it was triggered by the heating. The PMSE observed later (Area 2 and 3) are influenced by moderate precipitation. We considered:

- Area 1: high altitude weak PMSE observed around 20:30 UT at $88 - 90$ km

- Area 2: PMSE observed from 20:50 UT to 21:50 UT PMSE at $86 - 88$ km, in parts influenced by precipitation

- Area 3: from 22:00 UT PMSE influenced by moderate precipitation extending over altitudes $83.4 - 87.8$ km during about 30 heater cycles

**Observation 3: 05/06 August 2020.** During the third observation, PMSE were observed only before midnight. Some of the observations (Area 1 and 2) show no apparent influence of precipitation. Before the start of Area 1 there is PMSE present,

however this is not considered due to operational problems. For completeness we also consider Area 3 which displays a layered structure and is influenced by the heating. Both, the height and the shape suggest, however that this is not PMSE and rather sporadic E-layer. We considered:

- Area 1: strong PMSE in the absence of apparent precipitation for about one hour from 21:30 UT at $82 - 88$ km

- Area 2: PMSE at $83 - 87$ km in the absence of apparent precipitation between 22:50 UT and 23:50 UT

- Area 3: structure observed above 90 km from 22:45 UT consistent with a sporadic E-layer.

**Observation 4: 06/07 August 2020.** From the fourth observation we see a low altitude PMSE layer only slightly influenced by precipitation, a second layer at high altitude influenced by heating that also might be sporadic E-layer and a third area extending over a long period in time and many altitudes that does not seem to be influenced by particle precipitation. We considered:

- Area 1: a long interval of PMSE between $81 - 88$ km partly in the quiet ionosphere and partly influenced by electron
precipitation

- Area 2: Sporadic E-layer above PMSE height.

- Area 3: a weak PMSE with little apparent precipitation for about one hour from 21:30 UT at $82 - 88$ km

We find in general that the overshoot effect disappears in the presence of strong or moderate precipitation, as can be seen in the 15/16 August 2018 observation for example in Fig. 3. This is better illustrated in the figures given in Appendix A

where each area is enhanced. In the beginning of the observation campaign on 15/16 August 2018 (area 1), a weak PMSE developed under very quiet ionospheric conditions. We speculate that the heating possibly triggered the formation of PMSE in this instance. The echoes are only weakly enhanced in comparison to surrounding areas and the back-scattered power is reduced during heating and also an overshoot is observed (see Fig. A4 in Appendix A).





## 4 Observed PMSE modulation

First we summarize the heating effect and overshoots visible in all the observations and discuss these findings in context of previous observations. Then we discuss two selected cases, one with little or no particle precipitation and then one with moderate precipitation. Finally we compare a selected case with simulations of the overshoot cycle and discuss what information we can gain from modulating PMSE with heating.

### 4.1 Overall observational discussion

Here we summarize the decline and overshoot ratios from the histograms in Fig. D1-D4 in the appendix for all the four observations (3 areas each) shown in Table 2. In general the heating effect is seen in more than half of the heating cycles for each respective area with most of the average ratios showing values close to 0.75. It must be noted that these calculations show only the observations that have a value of $R_0 > 10^{10.5}$ to indicate the presence of PMSE and thus excluding most of the heating cycles that are not influenced by heating like random noise. This however causes the faintest PMSE to be excluded from the 225 histograms like is seen for the overshoot ratio for area 1 from 15 of August 2018, here the PMSE power is below the threshold and thus no cycles are included in the calculation despite 100 % of the cycles showing a decline due to heating. This would suggest that manually inspecting low power PMSE influenced by heating would be a better option or introducing other criteria to include these low power observations. This might be true for other observations as well and thus the threshold needs to be reconsidered, either the threshold for considering what is a PMSE is to high or that the heating influence can be seen to have 230 an effect on a layer that normally would not be considered as a PMSE since it most likely is at the threshold of detection for the radar being used and the irregularities are no longer comparable to the radar Bragg scale.

Two observations showed layers at very high altitudes and would not be considered PMSE, where their altitude range is more common for sporadic-E layers. This is Area 3 from 5-6 of August 2020, with altitude extending from around $91 - 98$ km, showing 3 distinct layers. And Area 2 from 6-7 of August 2020 (From this observations day there are two large layers of 235 possible sporadic E-layer, however we only focus on the area where clear heating can be seen). Examining the ratios in the table the heater seems to be effective in 66 % of the cases with about 0.75 mean value for those cases with visible heating for 5-6 of August and in 83 % of the cases for 6-7 of August with an average of 0.52 for those cases. This we would expect for a sporadic E-layer as it has been shown before that these types of layers can be influenced by heating (e.g Ignat'ev (1975) with a reduction in received power when the heater is on. For the overshoot ratio the histograms show only 10 % and 17 % of the 240 observations showing an increase after the heater is turned on and thus much more of the cycles are showing a decrease. We do not expect an overshoot to be present when heating a sporadic E-layer and these numbers confirm this. Also the average ratios of those few cycles that show an overshoot are very high, 0.87 and 0.89 respectively, indicating that background variation is more likely the cause for an increase and not an actual overshoot. The sporadic E-layers observed here are generally decreasing in amplitude with time and thus it stands to reason that in general the power received is smaller after the heater is turned off 245 than it was before it was turned on. Ignat'ev (1975) argued that when heating a sporadic E-layer the electron density present





in the layer would decrease and the layer would increase in thickness and thus spread over a larger area causing a reduction in power.

**Table 2.** Summary of histogram results (see Appendix D) for the decline ($R_1/R_0$) and the overshoot ($R_0/R_3$) ratio when they are smaller than 1 (indicating heating effect and overshoot) for all four observations. These numbers only include observations with minimum background amplitude $R_0 > 10^{10.5}$. A1 refers to area 1 for that observations date and so forth.

| | | Decline $R_1/R_0 < 1$ | | Overshoot $R_0/R_3 < 1$ | |
|---|---|---|---|---|---|
| | | Average of ratio | % of ratio | Average of ratio | % of ratio |
| 11/12.8.18 | A1 | 0.76 | 58 % | 0.56 | 45 % |
| | A2 | 0.75 | 61 % | 0.57 | 51 % |
| | A3 | 0.77 | 55 % | 0.61 | 50 % |
| 15/16.8.18 | A1 | 0.74 | 100 % | - | - |
| | A2 | 0.75 | 55 % | 0.64 | 31 % |
| | A3 | 0.69 | 63 % | 0.41 | 40 % |
| 5/6.8.20 | A1 | 0.72 | 46 % | 0.48 | 44 % |
| | A2 | 0.72 | 55 % | 0.44 | 10 % |
| | A3 | 0.75 | 66 % | 0.87 | 17 % |
| 6/7.8.20 | A1 | 0.74 | 59 % | 0.54 | 53 % |
| | A2 | 0.90 | 61 % | 0.89 | 10 % |
| | A3 | 0.52 | 83 % | 0.24 | 17 % |

To summarize; we see only overshoots in less than half of the cycles with many cycles often more influenced by background ionospheric conditions that might overshadow the heating of the PMSE. Ullah et al. (2019) show a larger occurrence of overshoots in their observations, with around 40-70 %, where their observations were during daytime. However in our case we see a few instances where the overshoot in some cycles are unusually large. Myrvang et al. (2021) found that a higher electron temperature due to heating could be achieved during night time compared to daytime which might help explain some of these large overshoots. However Kassa et al. (2005) found for their observations that the heating temperature effect observed increased for the observation with the most amount of sunlight (near noon).

Biggest obstacle in comparing with other observations done during midday are the different on/off times of the heater used in various observations as well as different criteria used in the data analysis and cycle averaging. Other possible reasons for





unusually large overshoots could be a change in the PMSE/NLC season as is noted by Latteck et al. (2021) that the season in getting longer and since our observations are at night and close to the end of the season there might be more varying background conditions influencing our observations than those during the day in June/July.

## 4.2 PMSE modulation under quiet ionospheric conditions

To discuss PMSE modulation under quiet ionospheric conditions, we choose an area where there is no apparent energetic particle precipitation seen; we consider the observations marked as area 2 in the Aug 05-06 2020 observation. A first inspection of the overshoot curves can be made using the overall power plot shown in Fig. 4. The beginnings of new heating cycles are marked with dashed lines when the heater is turned on. The dotted line indicates the time when the heater is turned off again. One can see that in many cases the PMSE signal changes both at the heater on and heater off time as well as during the cycles themselves. The PMSE layer lies within the altitude range 83-87 km with a maximum extension of 2 km at its widest. There are clear indications of a reduced PMSE power when the heater is on and in many cases we can see clear overshoots.

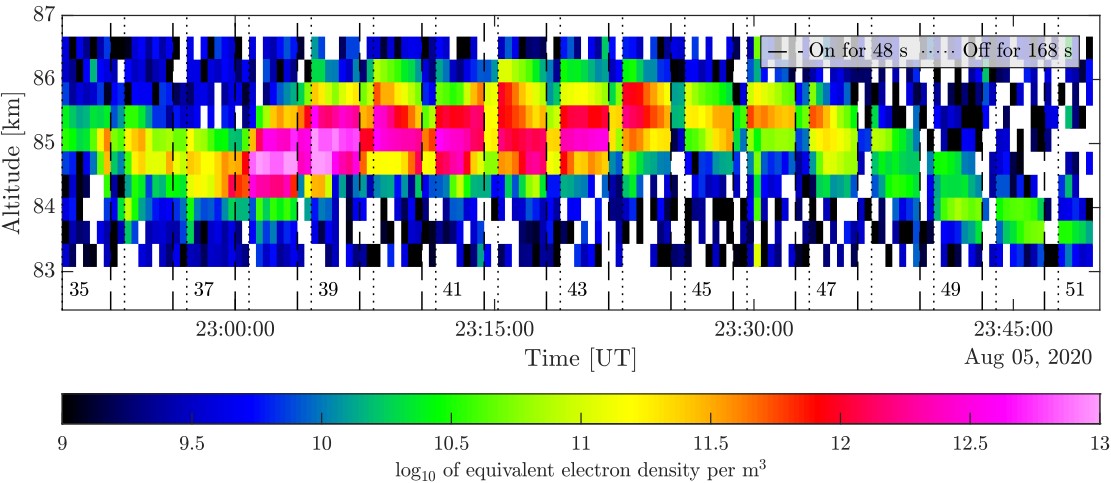

**Figure 4.** Back-scattered power as function of altitude and heating intervals observed during the night of August 05/06, 2020 in Area 2.

In Fig. 5 we have selected two altitude sections for a closer look, altitude 85.2 and 85.6 km, where we can clearly see overshoots in many of the cycles. In general the overshoots are rather large with some an order of magnitude larger than the pre-heater value and some showing no apparent increase in the PMSE power after heater turn-off. This seems to be especially true for the top altitude where the PMSE power is at its highest, the lower altitude has a somewhat lesser PMSE power and more overshoots are visible. The decline (due to the heater) is clearly visible in many of the cycles and is very strong for cycles 40-47. One can also see that characteristics of decline and overshoot especially often change between adjacent heating cycles and height intervals, as for example in heating cycle 41.



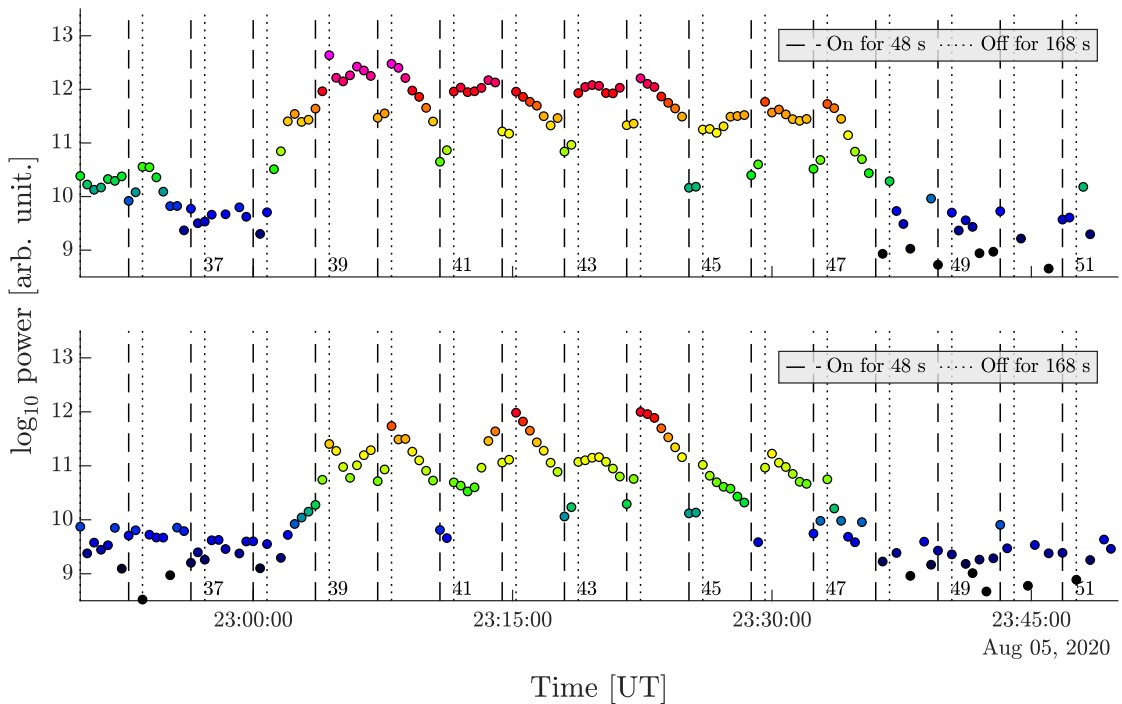

**Figure 5.** Back scattered power at altitude 85.2 (lower plot) and 85.6 km (upper plot) and heating intervals observed during the night of August 05/06, 2020 in area 2. The colour of the dots follow the colour scale of Fig. 4

For a closer investigation we describe the ratios of the amplitudes during the different phases of the heating cycle. The different power amplitudes are marked in the overall sketch given in Fig. 1: $R_0$ amplitude before heater on, $R_1$ amplitude after heater on, $R_2$ amplitude before heater off, $R_3$ amplitude after heater off, $R_4$ amplitude after decline. The $R_4$ corresponds to $R_0$ of the subsequent heater cycle. The different amplitudes observed during the heating cycles for the observation in question are plotted in Figure 6 where the amplitude ratios are considered. We find that during most heating cycles, the signal drops when the heating is switched on (decline $R_1 < R_0$, Fig. 6 a).

We assume that the observed signals are PMSE when they exceed a value of 10.5 in the given scale (logarithm of the equivalent electron density per $m^3$ which corresponds to around 3.16e+10 $m^3$) and one can see that in many cases that fall out of range of this condition there is no PMSE modulation seen, however as we will see later this condition removes a few cases of low power modulated PMSE as well. The same can be said for the green points that show a decline but are below the threshold that they could too be showing a decline but in fact be noise due to random fluctuations from the two measurement points.





**Figure 6.** Comparison of the power amplitudes observed on the 5/6 August 2020 in area 2.

The ratio of the amplitudes $R_0$ and $R_3$ describes the overshoot, when $R_0 < R_3$, and this comparison shows that overshoots and undershoots are equally abundant independent from the signal strength (Fig. 6b). Comparing the signals at the beginning





of subsequent cycles (Fig. 6c) shows no trend and a broad range of values which suggests variation either due to ionospheric
290    conditions or due to neutral turbulence (rather than dust).

One can see in Fig. 6d that for strong signals, the amplitude stays constant or decreases slightly during the heater on phase.
The change of amplitude during the heating can possibly indicate the charging process of the dust particles. Where the faster
timescale of diffusion or dust charging dominates (Mahmoudian et al., 2011). The comparison of $R_2$ and $R_3$ in Fig. 6e describes
to what extent the signal increase again when the heater is switched off. This increase is seen in most cases except for the small
295    amplitudes, these might be either very low power PMSE or random fluctuations.

Finally in Fig. 6f the ratio of $R_3$ and $R_4$ describes the signal after the heater is switched off again. Point below the diagonal
would describe the decline to normal conditions after an overshoot. Points close to the diagonal would describe that signals are
more constant in time. One can see a broad scatter symmetrically around the diagonal, indicating that the natural variations in
the PMSE power are dominant and any relaxation after heating is difficult to discern from this since their contribution could
300    disappear due to a large increase in PMSE power. This is due to the large time span between the two points (168 seconds),
which according to Havnes (2004) is enough time for the ionosphere to change or dust to become discharged where as 48
seconds used for the on time is not.

To quantify the results from Fig. 6 we compile the results in histograms of the values for the amplitude ratios. The histograms
contain only cycles that have a value $R_0 > 10^{10.5}$ of the PMSE amplitude before the heater is turned off to only include those
305    cycles that have PMSE and exclude the cycles that contain noise or are dominated by noise. We also only include those cycles
that show a decline due to heating in all the histograms. In Fig. 6a we see that 55 % of the ratios are smaller than 1, and thus
show a decline (affected by the heater) and that the average value of those ratios that are below 1 have a value of 0.72. So a
reduction of 28 % of the pre-heater value on average when the heater is turned on. In Fig. 6 b we have the overshoot and only
10 % of the cycles show an overshoot with an average value of 0.44 which even though there are not many overshoots for this
observation, those that are observed show an average reduction by more than a half, indicating very large overshoots. Fig. 6c
shows that most (95 %) of the observations show a decrease in power while the heater is on. Figure 6 d shows that 66 % of the
cycles show an increase in power when the heater turns off which is similar to the amount of cycles that show a reduction in
power when the heater is turned on. Then in Fig. 6e we see that there is a general increase in power from cycle to cycle and
thus a general decrease to pre-heater value can not be determined, most likely due to increasing background PMSE dominating
the signal and the histogram, where 87 % of the cycles show an increase in power in subsequent cycles. This can be related to
the reason we see so little overshoots in this observation, and that increase in PMSE power is large for many of the cycles and
the overshoot disappears due to background variations.





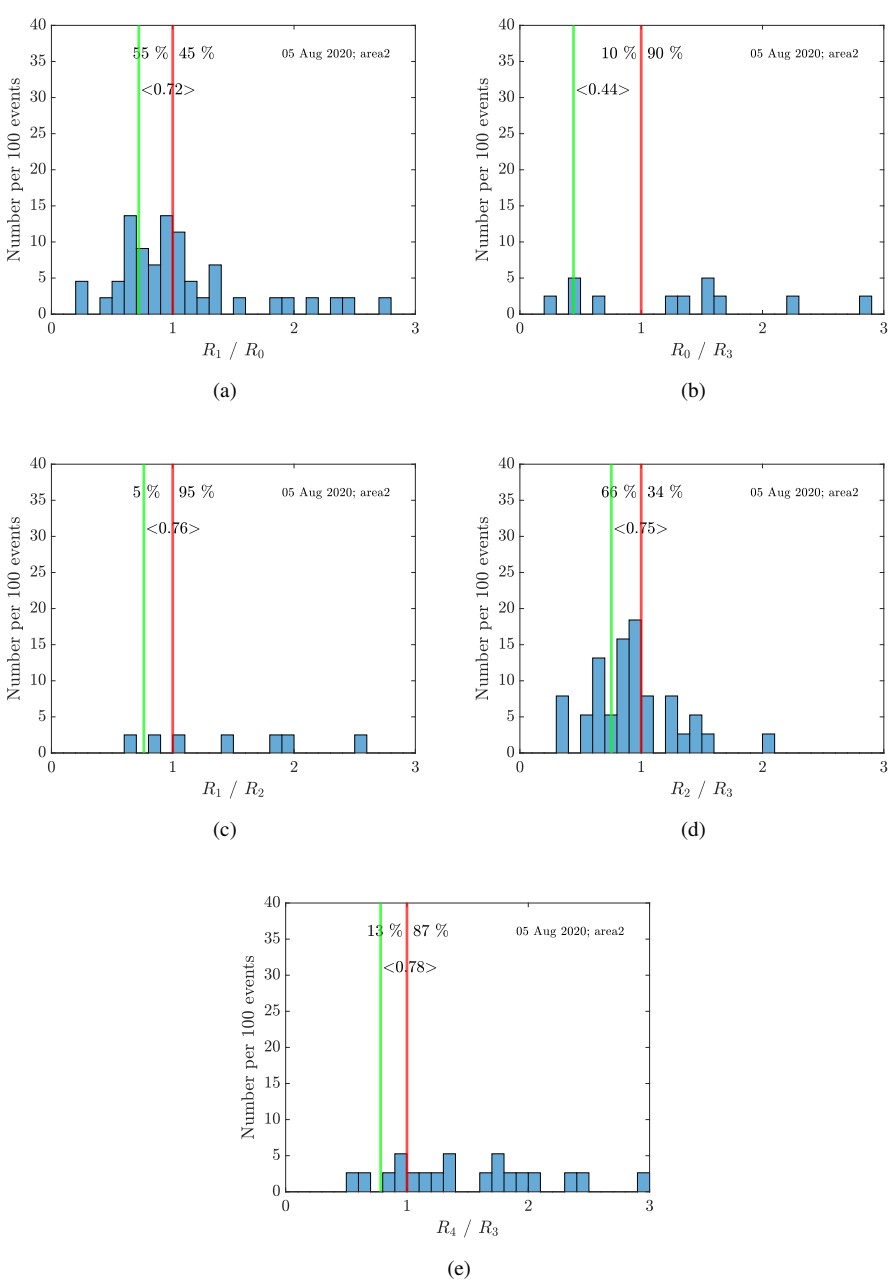

(a)

(b)

(c)

(d)

(e)

**Figure 7.** Average of (a) decline, (b) overshoot, (c) heating, (d) recovery, (e) relaxation for the observed data on the 5 August 2020 in area 2. Only overshoot curves with a minimal background amplitude of $R_0 > 10^{10.5}$ are considered. The ratios are chosen in such a way, that, if we observe an overshoot curve like shown in Figure 1, all ratios are smaller than 1. Thus, the histograms are clipped at a maximum ratio of 3. The green line and the corresponding number displays the mean for all ratios smaller than 1.





## 4.3 PMSE modulation during moderate particle precipitation

Conditions with moderate particle precipitation are observed in Area 2 of the observation made during the campaign on Aug
15/16, 2018. The overall power plot is shown in Figure 8. As in the other area previously discussed above, there are some
heating intervals with noticeably very strong overshoots (14,15,16,17). One can note the influence of the heating is most
pronounced in the beginning and the very end of the observation interval when there is no apparent particle precipitation.
Precipitation is clearly present in cycles 18 and 19 and then in cycles 24 and 25. Here when the heater is switched on there is
no reduction in power and the precipitation dominates the received signal for all altitudes in these cycles. The power plot for
two selected height intervals shown in Fig. B2 shows this in a more detailed way. Where the modulation fully disappears in the
cycles influenced by precipitation. This is to be expected and has been shown before, one of the reasons why the modulation
disappears in the PMSE layer is that the atmosphere below the layer is ionized due to the strong precipitation and the HF
radio wave might be strongly absorbed before it reaches the PMSE layer and thus not be strong enough to appreciably heat the
electrons in the layer.

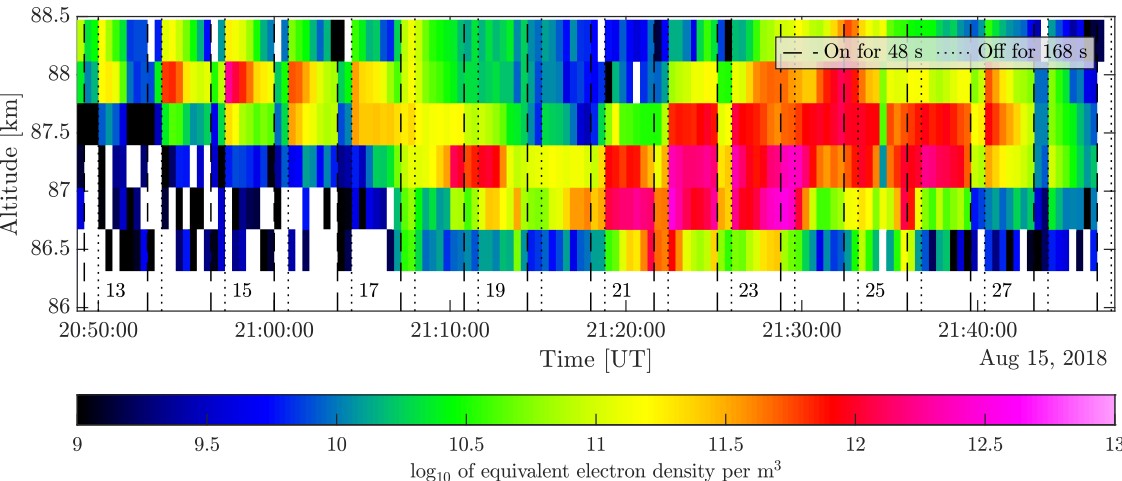

**Figure 8.** Back scattered power as function of altitude and heating intervals observed during the night of August 15/16, 2018 in Area 2.

The different amplitudes observed during the heating cycles in this area are plotted in Figure 10. We find that during most
heating cycles, the signal drops when the heating is switched on (decline, Fig. 10a). The cases that show no decline are spread
over all amplitudes, indicating the cycles that might be influenced by precipitation and thus might show an increase in power
when the heater is on. The overshoots and undershoots are equally abundant independent from the signal strength (Fig. 10b). As
observed in the area discussed above, there is no trend when comparing the signals at the beginning of subsequent cycles (Fig.
10c). The change of amplitude during the heating (Fig. 10d) is small for most of the observations. In most cases the amplitude
increases (Fig. 10e) when the heater is switched off similar to the cycles being heated which is to be expected. Finally in Fig.



(10f) the ratio of $R_3$ and $R_4$ describes the signal after the heater is switched off again, showing somewhat more of the cycles that are influenced by the heating showing a large spread around the diagonal with somewhat more observations showing a reduction. This large spread can be attributed to the ionospheric variability due to the large timescale of the off time as was 340 mentioned previously.

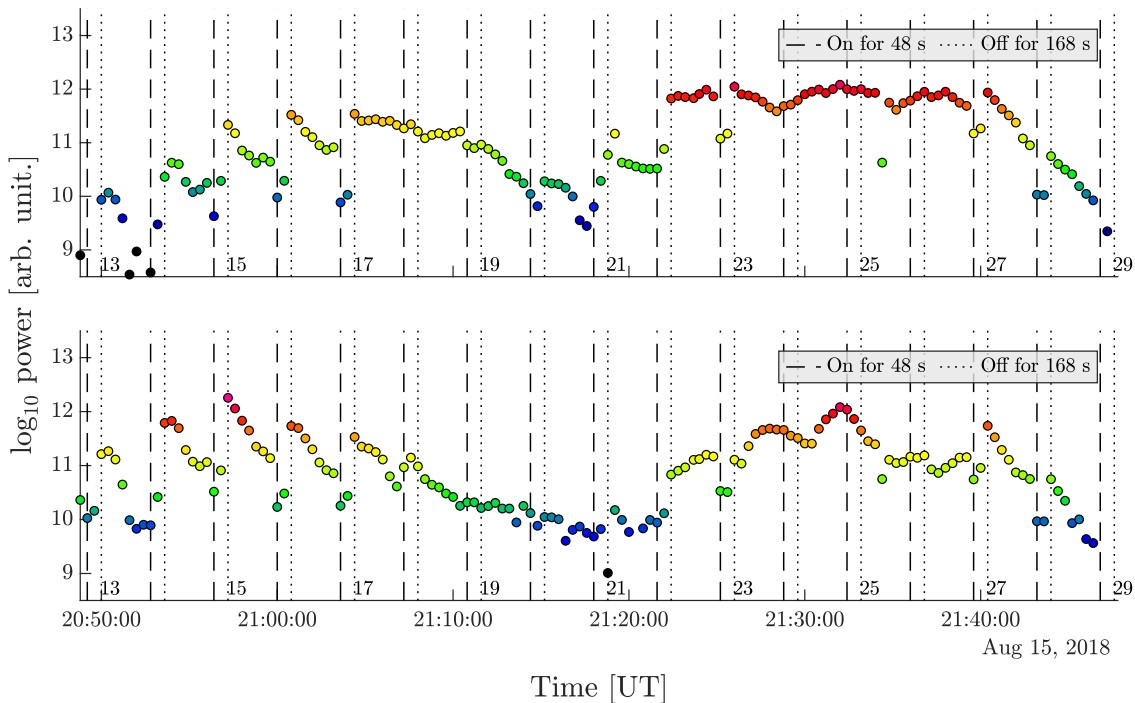

**Figure 9.** Back scattered power at altitude 87.4(lower panel) and 87.8(upper panel) km and heating intervals observed during the night of August 15/16, 2018 in area 2. The colour of the dots follow the colour scale of Fig8

The histograms of the power amplitudes are shown in Fig. 11 with the same criterion as before (also given in the figure text). Here the overshoot is seen in 55 % of the cycles with an average of 0.75 decline ratio (Fig. 11 a), similar to the previous observation. Here the overshoot is seen in 31 % of the observations with an average of 0.64 overshoot ratio (Fig. 11 b). Which is more than the previous observation, even with the presence of precipitation. Similar to the previous observation we see that 345 when the PMSE power increases in general (and is not influenced by precipitation) we see an influence of the heater but not an overshoot or a very small overshoot. For the cycles with a lower PMSE power (like in cycle 15) the overshoot is large but the background PMSE power is lower and thus the overshoot is easy to see. During the heating there seems to be a general decrease in the values with 76 % of the values showing a decrease during heater on (Fig. 11 c). The recovery (Fig. 11 d) ratio shows that 58 % has an increase in power when the heater is turned off. Showing similar values as for when the heater is turned





on (decline). Then there seems to be a little over half of the cycles that show a general increase in pre-heater values between cycles (Fig. 11 e).

**Figure 10.** Comparison of the power amplitudes observed on the 15 August 2018 in area 2.



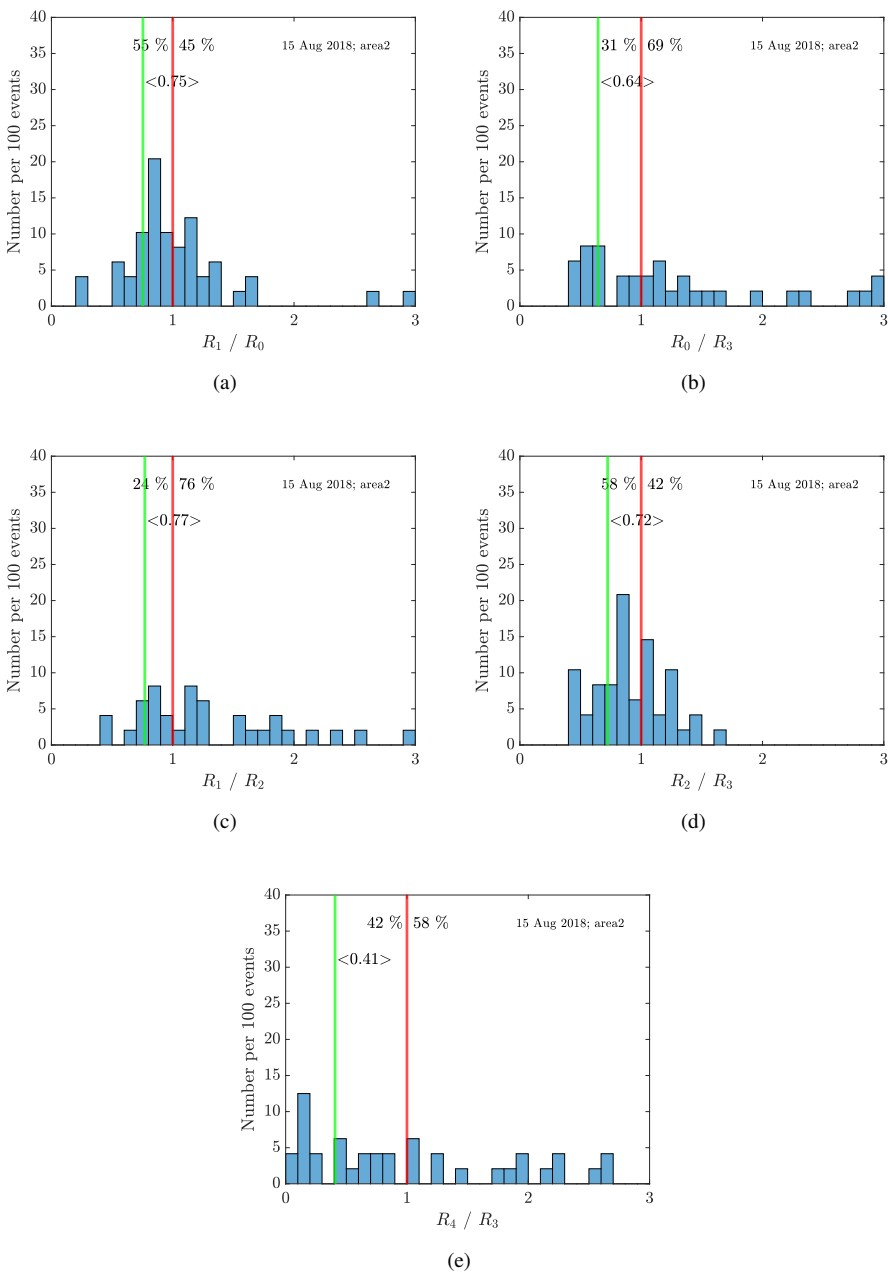

**Figure 11.** Average of (a) decline, (b) overshoot, (c) heating, (d) recovery, (e) relaxation for the observed data on the 15 August 2018 in area 2. Only overshoot curves with a minimal background amplitude of $R_0 > 10^{10.5}$ are considered. The ratios are chosen in such a way, that, if we observe an overshoot curve like shown in Figure 1, all ratios are smaller than 1. Thus, the histograms are clipped at a maximum ratio of 3. The green line and the corresponding number displays the mean for all ratios smaller than 1.





### 4.4 Comparison of a selected observation to simulation

Here we take a closer look at the approximate one-hour time interval which is marked as area 2 in the observation from 15-16

of August 2018, shown in Fig. 12; the data cover the heating cycles 12 to 27 and range over seven height intervals of around 360 m each. The ionosphere is influenced by precipitation in cycles 18 and 19 and then again in cycles 24 and 25, and there are no overshoots present in those heating cycles. We have also made this observation that the overshoots disappear during precipitation at other points in our measurements. The PMSE in intervals marked with A, B and C in the figure clearly respond to the heating. Interval A shows relatively low PMSE power but quite high overshoot curves in comparison to intervals B and C as we will now investigate further.

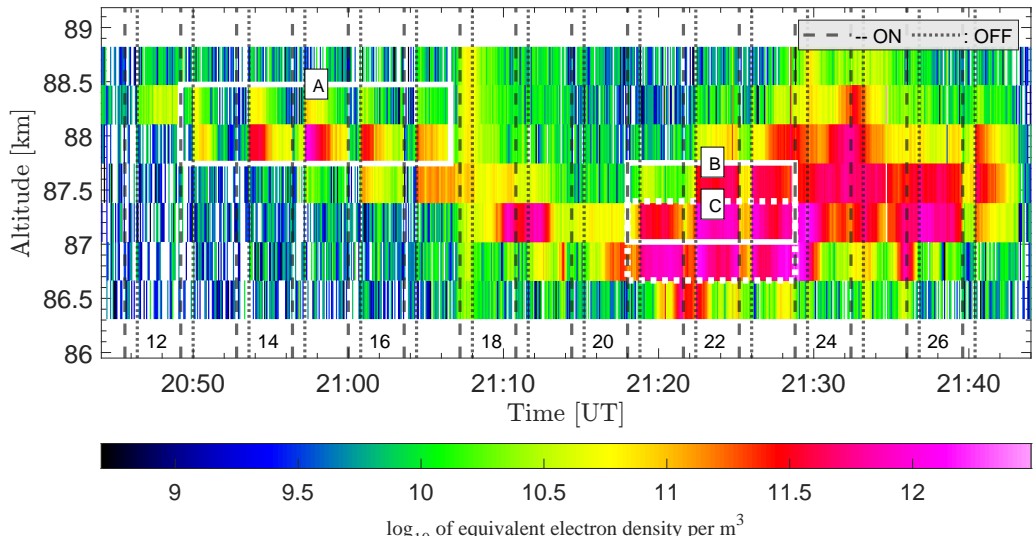

**Figure 12.** Overview of Area 2 - 15 of August 2018. With interesting visible overshoot cycles marked with intervals A, B and C. Data resolution is 4.8 seconds. Cycles are marked in the figure (from 12 to 27) as well as their corresponding On and OFF period

Individual heating cycles are shown in Fig. 13a for both altitudes from interval A, with PMSE power along with measurement error provided by the EISCAT GUISDAP analysis. Shown on the right in Fig. 13b in blue is the corresponding average overshoot cycle for the respective altitude. As can be seen the overshoot is rather strong for many of the heating cycles, especially strong overshoot is seen in cycle 15 for both altitudes with rather high but decreasing overshoot on both sides of the cycle. Note the two different y-axis scales for the different altitudes, where the heating cycles from altitude 88 km has

such a low background PMSE power that the scale is an order of magnitude lower than the altitude below. Both altitudes have relatively low background PMSE power compared to intervals B and C, where the cycles at 88 km altitude the PMSE is barely present or the irregularities on the limit of being seen by the VHF radar. It is thus interesting to find such large overshoot cycles for this particular interval.



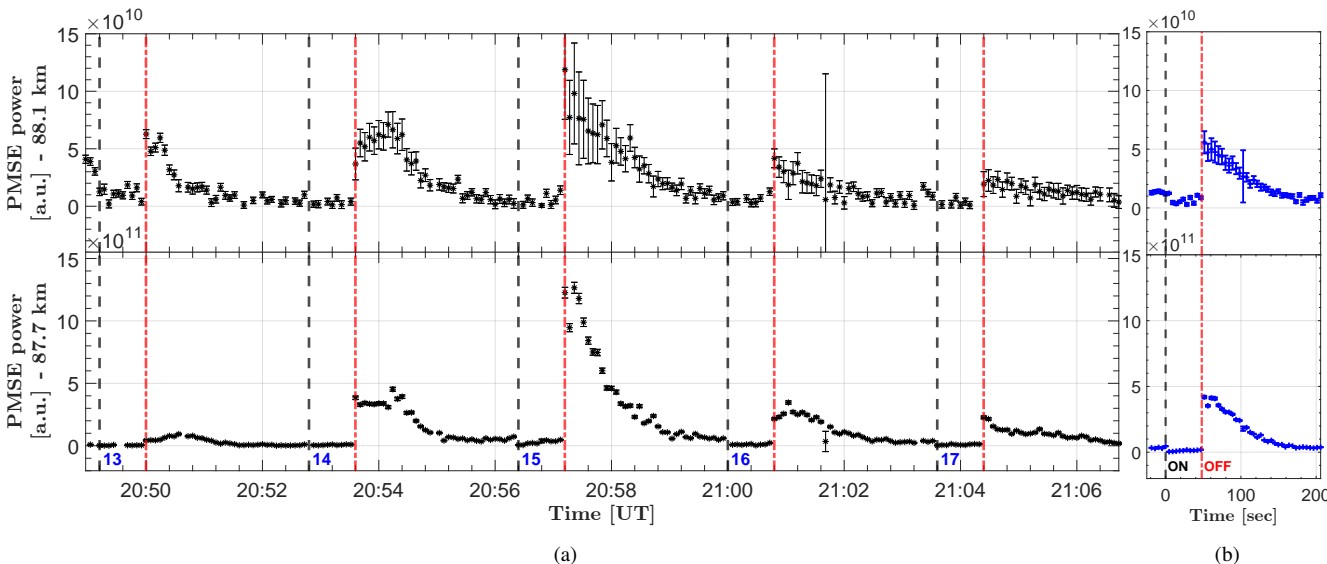

**Figure 13.** Individual overshoot curves (a) from interval A (from Fig. 12 shown with their corresponding altitude average on the right hand side (b). Heating cycle numbers are shown at the bottom as well as the on and off period for the averaged cycles. Note that the Y-axis scale for altitude 88 km is an order of magnitude smaller than for altitude 87.7 km.

Individual heating cycles from intervals B and C are shown in Fig. 14a with their corresponding altitude average on the right
hand side in blue (Fig. 14b) (Note that here the y-axis scale is the same for all the altitude ranges).They cover heating cycles 21, 22 and 23. For the lower altitudes the overshoots are present but are not as high as in interval A. However, the overshoot does not decline evenly but increases again before the initial signal level is reached. This influence can be clearly seen also in the averaged heating curve for altitude 86.7 km, where after about 120 seconds the power starts to increase again. We think that this is because of the beginning influence of particle precipitation on the ionosphere. This influence is very strong in the
subsequent cycle 24 where the PMSE power increases during the heater on period. This type of ionospheric variation can influence the observations to an extent that the heating effects are less visible. In the same time interval (intervals 21, 22, 23) at the altitude above, the overshoots are small, especially for the first cycle (21) while the PMSE power is also rather low. This is in contrast to the observation made at the higher altitude in interval A where a large overshoot is observed at low PMSE power. This might indicate that for these two cases there are different conditions at play.





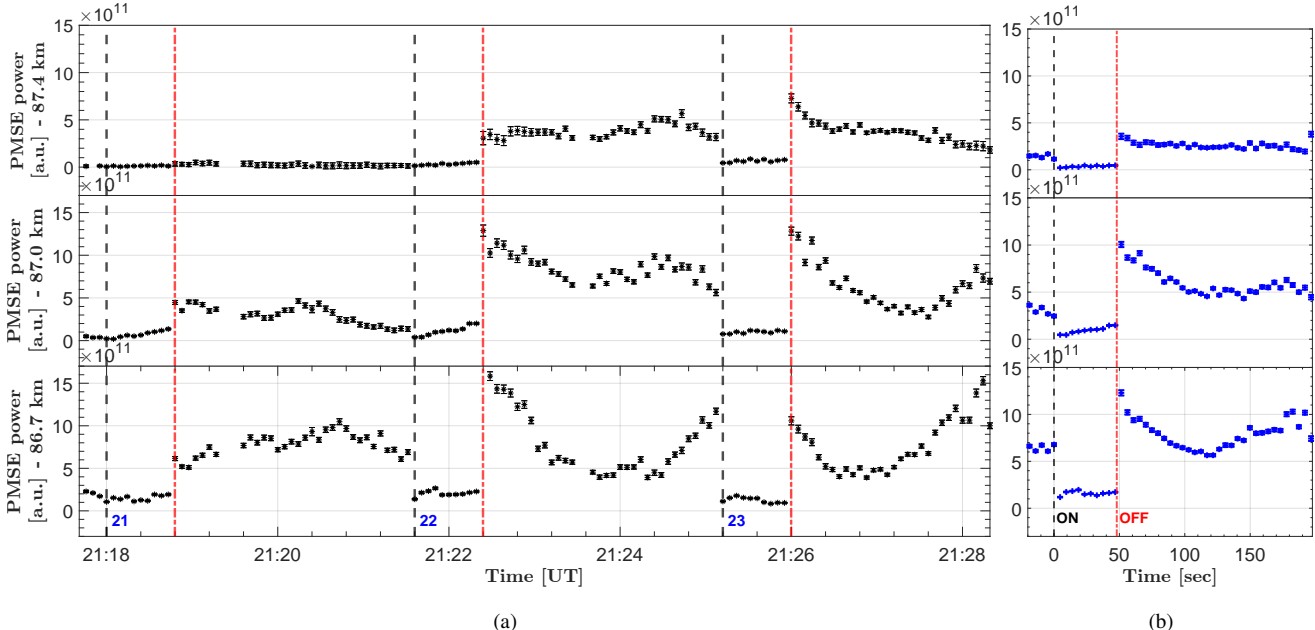

(a)                                                                      (b)

**Figure 14.** Individual overshoot curves (a) from interval B and C (from Fig. 12) shown with their corresponding altitude average on the right hand side (b). Heating cycle numbers are shown at the bottom as well as the on and off period for the averaged cycles.

Comparison of the average overshoot curves for each interval (A, B and C) is shown in Figure 15 on the left side and their corresponding normalized average curves on the right side. The values are normalized to the initial PMSE power taken as the average of the last 5 values (24s) before the heater is turned on. This is chosen so that we have sufficient data when some measurement points are missing and to better compare to the rest of the data used in this article which are at resolution of 24 seconds. Data were normalized after averaging the cycles from each interval. We can clearly see that the highest normalized overshoot (right) is the one from interval A which has the lowest background PMSE power (left) and that the lowest normalized overshoot is from interval C that has the corresponding highest PMSE background power. This high PMSE power is possibly due to an onset of precipitation which becomes apparent in the subsequent cycle 24 right after intervals B and C.



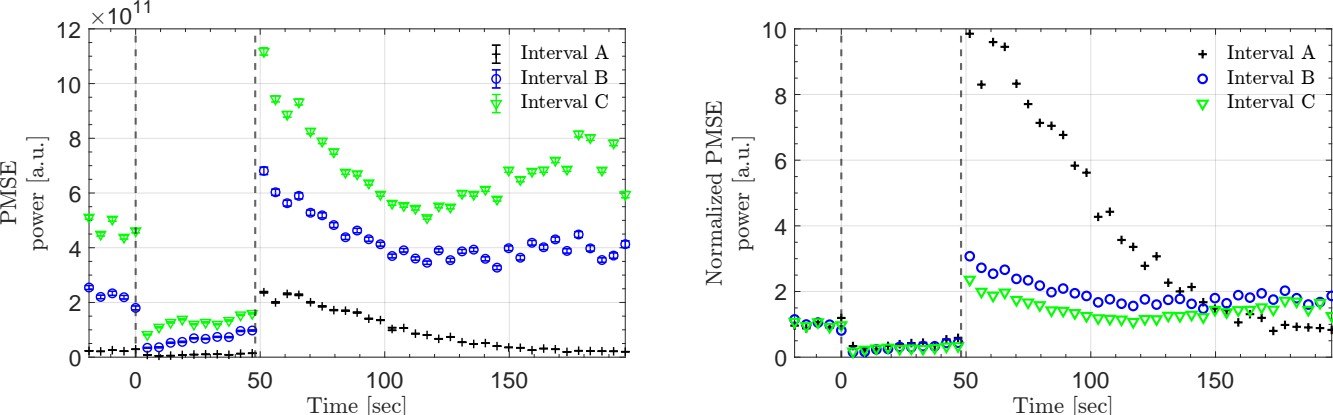

**Figure 15.** Average overshoot curves for each respective interval (right) and normalized average overshoot curves (left) for the same intervals. They are normalized with the average of the last 5 values before heater was turned on.

We compare these selected overshoot curves to a computational model which was initially developed at Virginia Tech. It treats the plasma as a fluid including an arbitrary number of charged particles, neutral particles and dust particles; the dust charging is described in the Orbital-Motion-Limited (OML) approach (see e.g Scales and Mahmoudian (2016)). The parameters of the model include the electron diffusion time scale, the charging time scale and the time evolution of electron and ion densities. The dust charging causes electron density depletion and the amplitude of electron density fluctuations determines the radar back-scattered amplitude. The simulations assume an initial plasma temperature of $T_i = 150\ K$ and a background electron density of $2 * 10^9\ m^{-3}$. Which fits well with the same parameters derived from the IRI-model (2016) for the time and date of the observation. The simulation also assumes a reduced photo-emission rate used in the charging equations in line with the experiments being done for conditions with low photo-emission.

The resulting simulated overshoot curves are shown on the right in Fig. 16 and for comparison are the averaged and normalized observations from intervals A, B and C (marked in same color and symbol as previous figures) shown on the left. The simulations show best fit to the observed overshoot curves for 3 nm dust particles. However there is little difference for similar sizes of dust (e.g 3-4 nm). This result fits well with the altitude range we measure the observed PMSE echoes since in general we can assume to find smaller particles of dust at higher altitudes (however subject to neutral air movement) as well as the fact that there were no NLCs observed and thus the particles were not optically visible (larger >20 nm).

The normalised and averaged data from interval A has a higher overshoot than what the simulations can produce, where the simulation has an overshoot of around 8.4 while the observations show an overshoot of almost 9.9. Note also the timescale of the simulation for interval A runs for 300 seconds while the observation has a much quicker equalization towards the "background" PMSE value/undisturbed plasma values. For the simulation to reach such a high overshoot the ratio between dust and electron number density is only at 35 % and with a heating ratio increase for electron temperature of 8 times the pre-heater value. This would indicate that the dust density is lower than for the other two intervals and that the effect of the

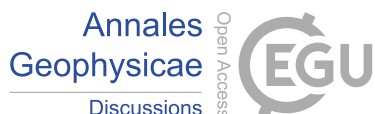

heating is consequently larger. The electrons gain a higher temperature and charging onto the dust particles is consequently
more effective. Where some dust particles can gain more than a single charge as is discussed later.

Comparison of observations for intervals B and C and their corresponding simulations show a better "match" where the
overshoot and relaxation are very similar. For these overshoots to be produced in the simulation the ratio of dust to electrons is
higher with 60 % for interval B and 68 % for interval C. The increase at the end of the relaxation period for both intervals is not
reproduced in the simulations, this is assumed to be due to the influence of the precipitation that occurs clearly in cycle 24 and
already is increasing the background PMSE power in the previous cycles. Compared to the observations, the simulated signals
drop more slowly during the heater ON phase and rise more slowly to the overshoot when the heater is switched off again.
The measured response of the PMSE to the heating is instantaneous within the 4.8 seconds resolution of the data. A possible
explanation for this difference is that the numerical model might have missing parameters or processes that are present in order
to simulate this increase. This is in contrast to the decrease we see in most of the observations as was discussed previously.

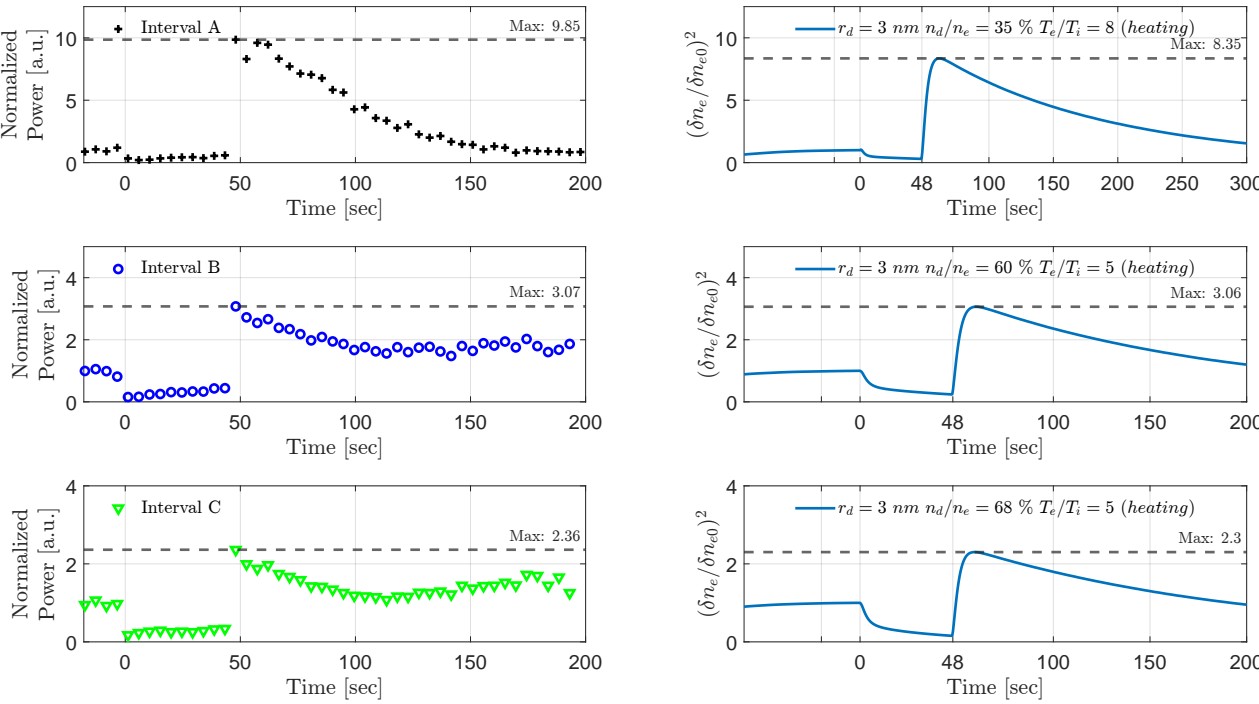

**Figure 16.** Comparison of the averaged and normalized heating cycles for each interval (left) to its corresponding simulation of the overshoot
cycles (right).

On the left hand side in Fig. 17a we can see the average charge number found for the simulation for each respective interval
(marked in the figure). For interval A the average charge number reaches a maximum of about 1.38 charges per dust particle
during the heater on phase. This indicates that in order to achieve such a high overshoot the charging efficiency of the dust
particles needs to be high and that (due to high electron heating temperature) many dust particles will gain more than one
negative charge during the heating cycle. Note the longer timescale shown in the simulation for interval A (300 seconds),





indicating that it takes a longer time for the overall average charge on the dust particles to equalize back to pre-heater values. This is also true for the simulation of the timescale for interval A (Fig. 17b) where the timescale goes to 300 seconds but is cut at 200 seconds to better compare to the simulations for the other intervals. As was mentioned before the dust population seems to be much lower for interval A compared to the other two intervals since the ratio of dust to electrons is lower and consequently the large increase in temperature (by a factor of 8) causes a larger average charge number on the dust particles

during the heater on phase. For the other intervals (B and C) the maximum average charge number is less than one during the on phase of the heater for both cases, with interval B being around 0.9 charges per dust and for interval C the average dust charge lies around 0.86. Which corresponds well with the observed and simulated overshoot curves from Fig. 16 where the higher overshoot is observed in interval B and thus the average charge number is consequently higher. So the effective charging of the dust during heater on for these intervals is less than for interval A and a smaller overshoot is observed.

On the right hand side in Fig. 17b we have the ratio of the diffusion time to the charging time scales for each respective interval. Here we can see the variation between the two timescales and how this changes during the heating cycle. For all the intervals there is an increase in the ratio when the heater turns on, a relaxation during the heater on period and a short increase when the heater is turned off and a slow decrease during the heater off period. The large increase at heater on time could be understood in terms of the charging timescale becoming smaller with increased electrons charging onto dust particles due to

the increased electron temperature. This corresponds well with the increased average electrons charge on the dust particles seen in Fig. 17a. Here the average dust charge is highest for interval A and the ratio of timescales is also highest for this interval. Which might indicate a faster charging timescale for that interval than for the other two. The increase at heater turn off is then also due to a decrease in the charging times, more dust is being charged now by the ion portion of the plasma which drag the electrons along and cause the observed overshoot. Thus for interval A the simulation of the overshoot curve fits best with a

lower ratio of dust particles to electron density and thus we might argue that there is more plasma compared to the other two intervals. This larger plasma population might then charge the dust more and more quickly cause a smaller charging timescale and consequently a larger overshoot in interval A.



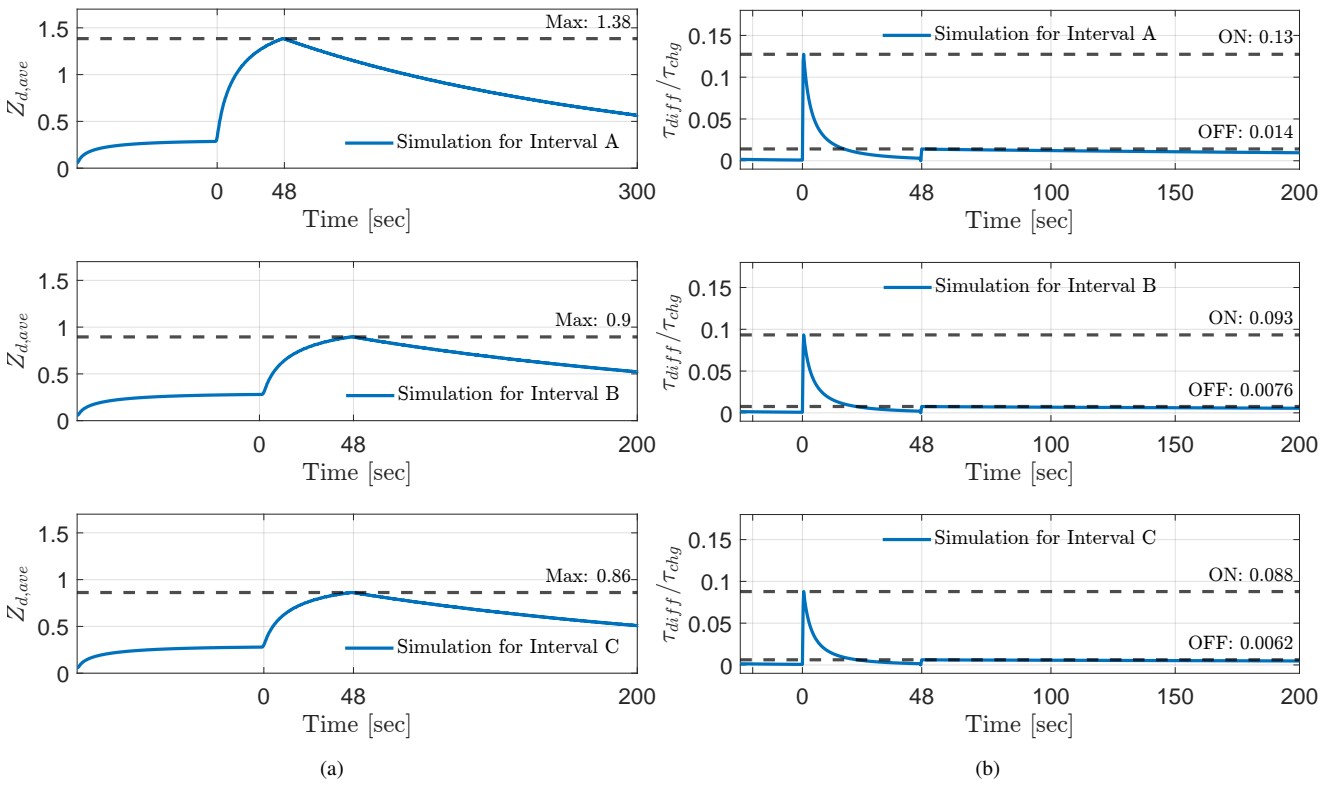

**Figure 17.** Simulations of average dust charge number on the left (a) for each respective interval and the ratio between the diffusion time and the charging time scales for same intervals on the right (b).

Another influence also worth noting is the possible difference in the diffusion timescales of the respective intervals before the heater is turned off. This difference could arise due to the higher altitude of interval A and consequently a lower neutral
density. This small reduction in neutral density causes a smaller ion-neutral collision frequency, and consequently a smaller diffusion time scale due to the diffusion being proportional to the collision frequency. Where the diffusion timescale can be estimated with the following equation (Chen and Scales, 2005):

$$\tau_{dff} \approx \nu_{in} \left(\frac{\lambda_{irreg}}{2\pi v_{th}}\right)^2 \frac{1}{(1 + \frac{T_e}{T_i})(1 + \frac{z_d n_{d0}}{n_{e0}})} \tag{1}$$

Where $\nu_{in}$ is the ion-neutral collision frequency, $\lambda_{irreg}$ is the wavelength of the irregularity and is given by $2 * \lambda_R$ (the
radar wavelength), $v_{th}$ is the mean ion thermal velocity, $T_e$ and $T_i$ are the electron and ion temperatures respectively. The dust charge number is given by $Z_d$, $n_d$ is the dust number density and $n_e$ is the background electron number density. The ion-neutral





collision frequency $\nu_{in}$ (polarization) can be estimated using the following equation ((Ieda, 2020; Cho et al., 1998) see ref. for relevant constants used in the equation):

$$\nu_{in} = 25.879 \times 10^{-16} \frac{N_n}{\sqrt{m_i}} \sum_t F_t \sqrt{\frac{m_n \alpha_0}{m_i + m_n}} \tag{2}$$

Where $N_n$ is the neutral density (in $m^{-3}$), $\alpha_0$ is the neutral gas polarizability ($10^{-24}$ $cm^3$), $m_i$ and $m_n$ are the ion and neutral mass (in atomic mass units - amu) respectively. The sum can be summed over the fractional volume of the neutral gas present, $F_t$, of each respective neutral constituent (here used most abundant; molecular nitrogen, oxygen and atomic argon). A quick estimate of the diffusion timescale right before the heater is turned on and right before it is turned off using values for the neutral density (from NRLMSISE-00 Atmosphere Model (Hedin, 1991)) is shown in Table 3 below. Here we can see

that the estimated timescale of the diffusion for interval A is lower than for the other two intervals for both diffusion estimates, indicating that a smaller charging timescale is needed to get the timescale ratio shown in Fig. 17b an thus a much more effective charging must be present for interval A.

Table 3. Neutral density for each interval from NRLMSISE-00 Atmosphere Model) (Hedin, 1991) taken at 21 UT and the estimated ion-neutral collision frequency (Eq. 2) and a rough estimate of the diffusion timescale (Eq. 1 right before heater is turned on and right before the heater is turned off

| Interval | Neutral density $[m^{-3}]$ | $v_{in}$ [1/s] | $\tau_{dff}$ at 0 sec | $\tau_{dff}$ at 48 sec |
|---|---|---|---|---|
| A | $1.19 * 10^{20}$ | $3.44 * 10^4$ | 0.0256 | 0.0188 |
| B | $1.33 * 10^{20}$ | $3.85 * 10^4$ | 0.0407 | 0.0304 |
| C | $1.48 * 10^{20}$ | $4.26 * 10^4$ | 0.0443 | 0.0327 |

The timescale that is the fastest is the dominating one. So when the heater turns on the diffusion timescale might be lower for interval A. So when the heater is turned on the diffusion timescale decreases even more due to its dependence on the

temperature ratio (Te/Ti) and we expect/need a larger temperature increase for the electrons in interval A to explain such a large overshoot. As the heater is turned on the charging timescale decreases due to the increase in electron temperature and a larger charging effect is seen in the interval A simulation (average charge number) compared to the other intervals and consequently a larger overshoot is seen. So to summarize it might appear that the decreased diffusion timescale for interval A due to decreased neutral density and the large increase in electron temperature combined could help explain the large overshoot

that is seen for interval A. However where the effect of the heater is absorbed and how much must have something to say. If some amount of the heater effect is absorbed below the PMSE layer due to increased ionization it could influence the observed overshoot. And same for layers that absorb much of the incoming heater radio wave could cause a larger overshoot.



## 5 Discussion and Conclusion

For the presented observations we find that the artificial heating affects the signals during less than half of all the observed
heating cycles with a pre-heated PMSE power $R_0 > 10^{10.5}$; the average reduction of the power is about 25 % from the pre-
heated value. The cutoff, $R_0 > 10^{10.5}$, excludes cycles that are not showing PMSE and/or cycles being highly influenced by
noise. With this criterion we covered most of the PMSE, however some very faint ones were excluded, some of them were
clearly affected by heating and showed an overshoot. We find that the heating has little effect on PMSE during ionospheric
conditions with particle precipitation which is also seen by other authors. This is especially so for strong and moderate particle
precipitation. We assume that under these conditions of higher ionisation the heating waves are mainly absorbed at lower
altitudes, thus not causing a heating effect in the PMSE layer. Often the background ionospheric conditions strongly influence
the PMSE profile during one heater cycle and it is particularly difficult to derive the relaxation time, which would be an
interesting parameter because it depends on the dust conditions present in the layer. Typically the overshoot is not so strong
when the PMSE power increases as a result of increasing ionization.

As to the shape of the PMSE modulation curves, the variation of the PMSE during the heater-on period(from $R_1$ to $R_2$)
is affected by two competing processes: the charging and the diffusion. For the presented observations most of the observed
heating cycles display a signal decrease from R1 to R2. Less than half or the cycles that are influenced by heating display an
overshoot when the heater is turned off. However, observed overshoots are generally high and in some cases very high. These
high overshoots could be attributed to the fact that the dust charge in the presented observations is more strongly influenced
by heating, as the influence of photoemission is smaller than during day-time observations. It is also possible that the size of
ice particles and their formation and sublimation rates are different at this time toward the end of the PMSE season; most other
heating studies were carried out earlier in the year. A general trend towards a longer PMSE season (Latteck et al., 2021) and
larger particles at PMSE altitudes (at high latitudes) due to increased water vapor content (Lübken et al., 2021) could also cause
these recent PMSE observations to show different modulation curves.

The heating model we considered cannot account for some of the very high overshoot cases we observed and we are not
aware of a model that does so, there might be some processes that need to be included to reproduce these special cases. The
modulations, i.e. the overshoot curves are often studied considering average curves including several heating cycles. We find
that the influence of variation in the ionospheric background with time over the cycles reduces the overshoots and dominates the
relaxation phase. We form however averaged curves as was done in other studies and compare those to the model calculations.
We find that simulations assuming dust size around 3 nm fits best to all cases considered; while different electron heating ratios
and different ratios of dust to electron density are needed to match the observational data, with a larger heating ratio and a
lower dust density is needed to best match the large average overshoot seen.

Finally, the heating itself is an unknown. Since the electron temperature influences the charging processes observed in the
heating cycles it is important to correctly estimate the electron temperature during the heating. Recently, Myrvang et al. (2021)
showed that the presence of dust particles affects the HF heating and the resulting electron temperature as a function of altitude.
This implies that when estimating the electron temperature, one needs to make assumptions on the dust components below the



PMSE. In addition the electron density below the PMSE plays a role. This might be the reason why we see large overshoots for example in the high altitude layer of Area 2 from 15 of August 2018 where there is no precipitation before the layer starts and the PMSE power is relatively low. Thus the heating might be quite effective in this case than for the later conditions where precipitation causes an overall higher electron density and therefore also absorption of the heater waves already below the PMSE.

The presented observations during HF heating confirm that PMSE amplitudes are modulated by high-power radio waves with the observed modulation varying on short spatial and temporal scales. Presented observations differ from previous studies since they are done late in the PMSE season as well as during lower solar illumination (dusk/night). In general we see both an influence of the heater as well as an overshoot in about half of the heating cycles. We see cases of very high overshoots compared to other previous observations. However, due to the large background and ionospheric variability, any specific determination of dust parameters or conditions causing these large overshoots is difficult as there are many unknown parameters related to PMSE heating. These include formation and disappearance of ice particles, horizontal and vertical movement of the layer, active turbulence, dust charge and size distributions within the PMSE layer and in the layer where the effect of the heater is mainly absorbed.

*Code and data availability.* EISCAT VHF data used is available under https://madrigal.eiscat.se/madrigal. EISCAT GUISDAP analysis tool used is available under https://eiscat.se/scientist/user-documentation/guisdap/. Information on the Manda radar code is available under https://eiscat.se/wp-content/uploads/2021/03/Experiments_v20210302.pd. Code and plots used in this article can be accessed at the UIT data repository; https://doi.org/10.18710/NGISOA

*Author contributions.* TG, AP and IM did the data analysis and interpretation. Planning of the observations was done by IM, MR, IH and PD. Temperature data from AURA satellite provided by PD. Investigation of overshoot done by TG and AM and simulations and model provided by AM. Everybody participated in manuscript writing, preparation and interpretation.

*Competing interests.* The authors declare that they have no conflict of interest. At least one of the (co-)authors is a member of the editorial board of Annales Geophysicae.

*Acknowledgements.* This work is carried out within a project funded by Research Council of Norway, NFR 275503. The Norwegian EISCAT participation is funded by The Research Council of Norway project 245683. The publication charges for this article have been funded by a grant from the publication fund of UiT The Arctic University of Norway. The EISCAT International Association is supported by research organizations in Norway (NFR), Sweden (VR), Finland (SA), Japan (NIPR), China (CRIRP), and the United Kingdom (NERC).

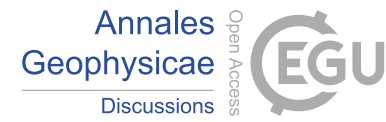

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





Additional figures depicting the observations are given in Appendix A, with selected cases discussed in more detail in Appendix B. Appendix C then contains a comparison of the power amplitudes for each respective measurement range and Appendix D contains histograms of these amplitudes.

**Appendix A: Overview of the measurements**

**Table A1.** Days of measurements and selected areas. $t_{start}$ and $t_{end}$ define the beginning and the end of the area. The altitude of the atmosphere, where the analysis is done, are described with $h_{low}$ and $h_{high}$.

| Day / Area | $t_{start}$ | $t_{end}$ | $h_{low}$ | $h_{high}$ |
|---|---|---|---|---|
| Night of Aug 11/12, 2018 | Aug 11, 20:00 | Aug 12, 02:00 | 80.0 km | 110.0 km |
| Area 1 | Aug 11, 21:36 | Aug 11, 22:42 | 83.4 km | 85.6 km |
| Area 2 | Aug 11, 23:06 | Aug 12, 01:17 | 86.3 km | 90.0 km |
| Area 3 | Aug 12, 00:00 | Aug 12, 01:28 | 83.4 km | 86.4 km |
| Night of Aug 15/16, 2018 | Aug 15, 20:00 | Aug 16, 02:00 | 80.0 km | 110.0 km |
| Area 1 | Aug 15, 20:06 | Aug 15, 20:25 | 88.1 km | 89.6 km |
| Area 2 | Aug 15, 20:48 | Aug 15, 21:47 | 86.3 km | 88.5 km |
| Area 3 | Aug 15, 21:57 | Aug 15, 22:59 | 83.4 km | 87.8 km |
| Night of Aug 05/06, 2020 | Aug 05, 20:25 | Aug 06, 00:00 | 80.0 km | 110.0 km |
| Area 1 | Aug 05, 21:25 | Aug 11, 22:50 | 82.0 km | 88.0 km |
| Area 2 | Aug 05, 22:50 | Aug 12, 23:50 | 83.0 km | 87.0 km |
| Area 3 | Aug 05, 22:45 | Aug 06, 00:00 | 90.0 km | 100.0 km |
| Night of Aug 06/07, 2020 | Aug 06, 21:15 | Aug 07, 02:00 | 80.0 km | 110.0 km |
| Area 1 | Aug 06, 22:53 | Aug 07, 02:00 | 81.5 km | 88.0 km |
| Area 2 | Aug 06, 22:43 | Aug 06, 23:29 | 91.0 km | 94.0 km |
| Area 3 | Aug 06, 21:15 | Aug 06, 22:15 | 82.0 km | 85.0 km |





**Table A2.** Values of ERP given in the EISCAT Heating facility logs from sample beam patterns for each of the measurements for reference. It seems that on 6 Aug 2020 at around 23:08:25 UT three transmitters changed phases such that the beam became broader, with about 360 MW X-mode and 17 MW O-mode which remained so until the end, which is why we have 359 MW X-mode at 01:07:13 UT (7 of August) below) 359 at 7

| Day | Time (UT) | ERP |
|---|---|---|
| 11 August 2018 | 20:50:13 | 560 MW |
| 12 August 2018 | 01:20:13 | 541 MW |
| 15 August 2018 | 20:06:19 | 568 MW |
| 16 August 2018 | 00:39:49 | 580 MW |
| 05 August 2020 | 20:47:01 | 495 MW |
| 06 August 2020 | 19:29:58 | 567 MW |
| 7 August 2020 | 01:07:13 | 359 MW |

## A1 Night of Aug 11/12, 2018

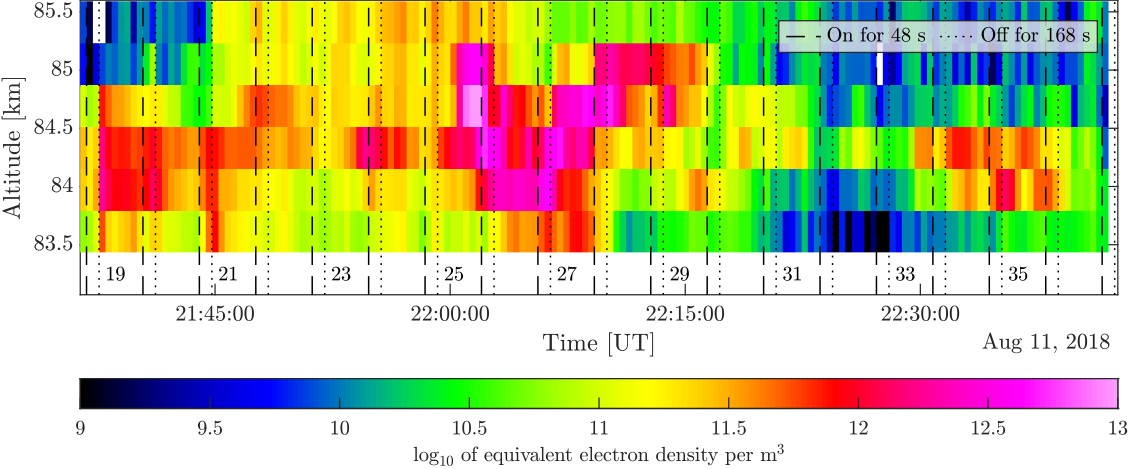

**Figure A1.** Back scattered power as function of altitude and heating intervals observed during the night of August 11/12, 2018 in Area 1.

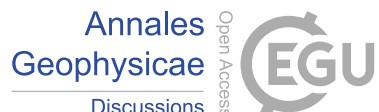

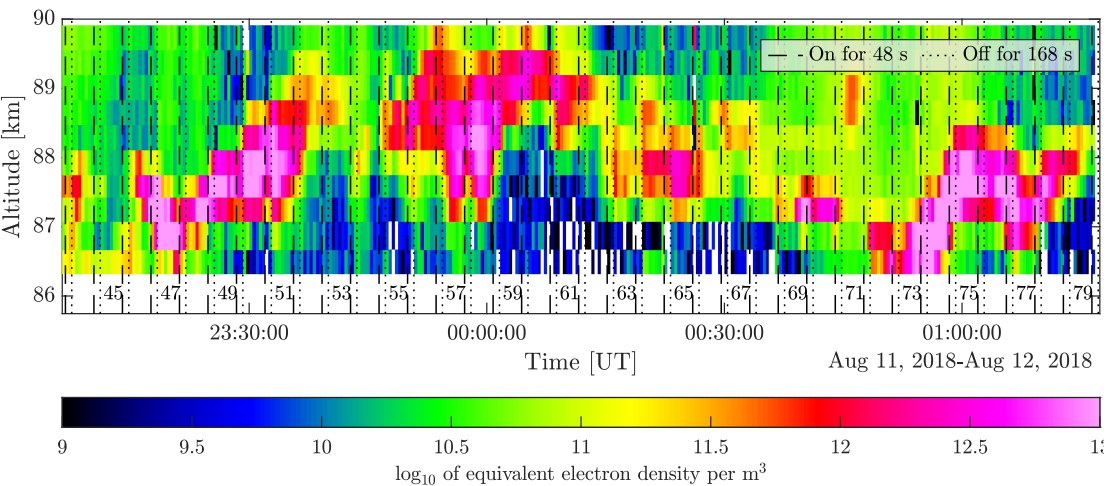

**Figure A2.** Back scattered power as function of altitude and heating intervals observed during the night of August 11/12, 2018 in Area 2.

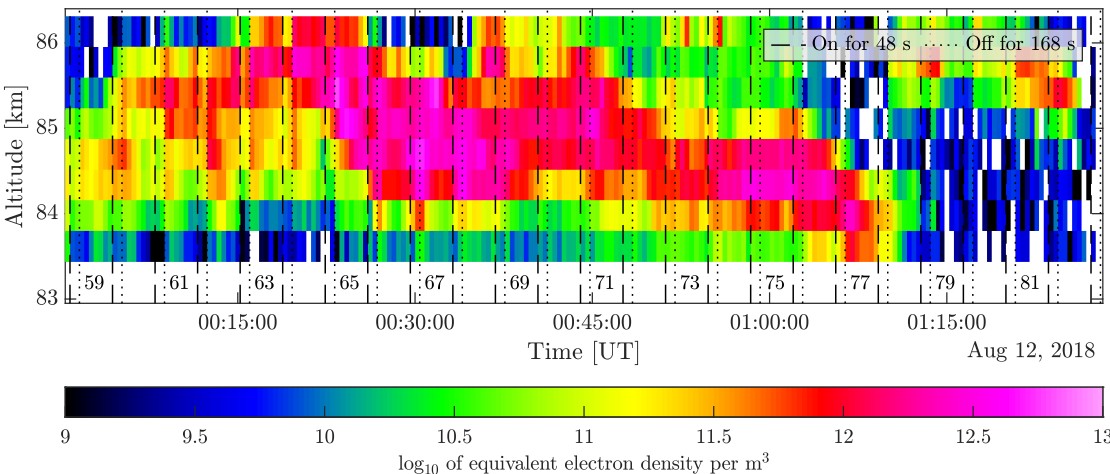

**Figure A3.** Back scattered power as function of altitude and heating intervals observed during the night of August 11/12, 2018 in Area 3.

## A1 Night of Aug 15/16, 2018

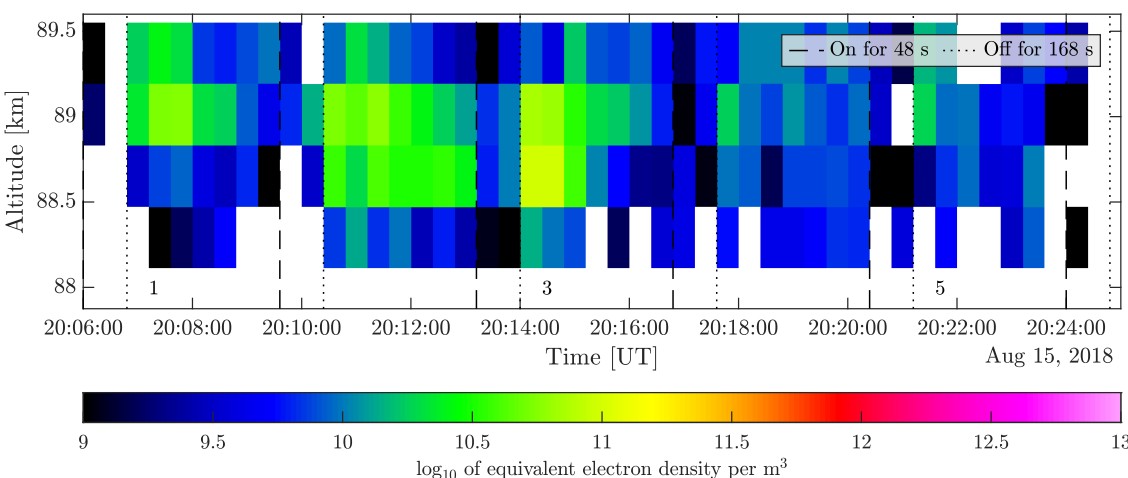

**Figure A4.** Back scattered power as function of altitude and heating intervals observed during the night of August 15/16, 2018 in Area 1.

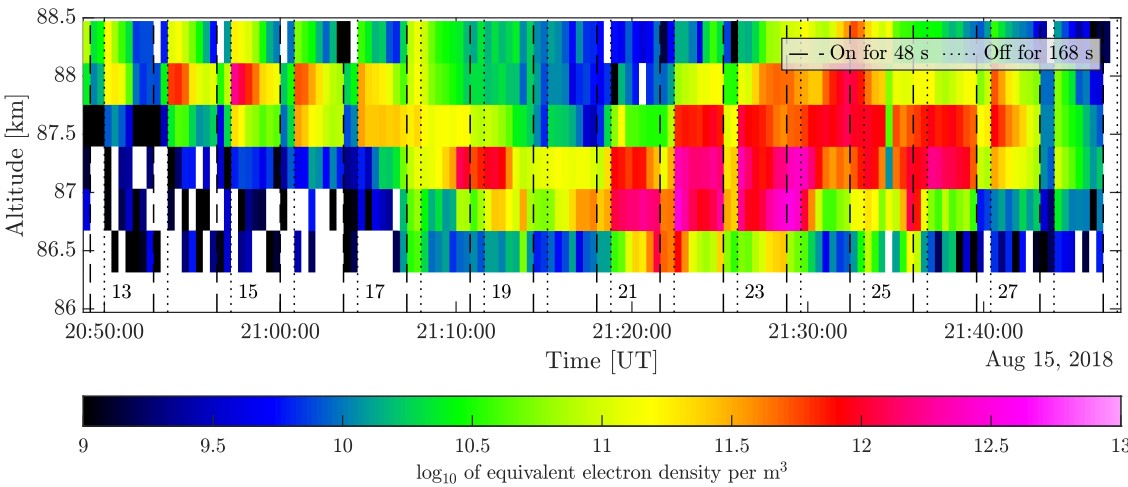

**Figure A5.** Back scattered power as function of altitude and heating intervals observed during the night of August 15/16, 2018 in Area 2.




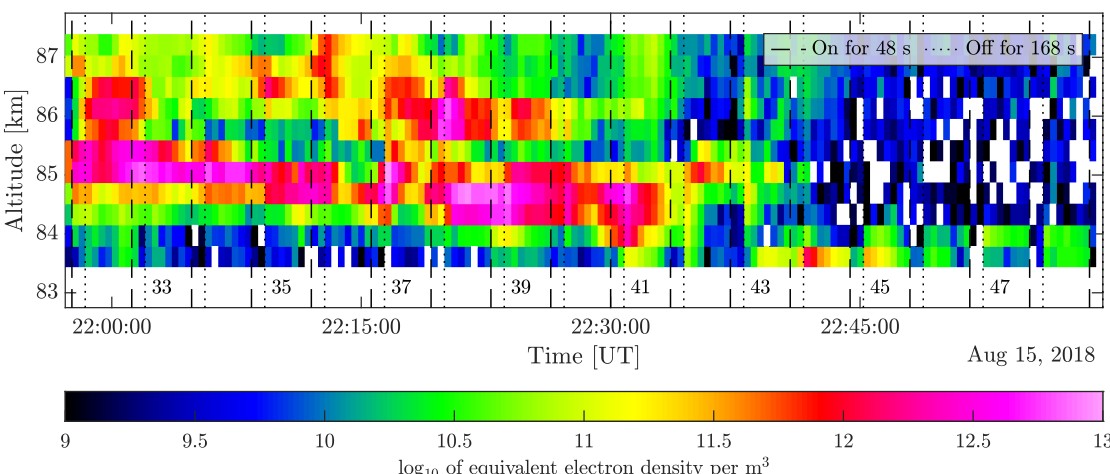

**Figure A6.** Back scattered power as function of altitude and heating intervals observed during the night of August 15/16, 2018 in Area 3.

## A2 Night of Aug 05/06, 2020

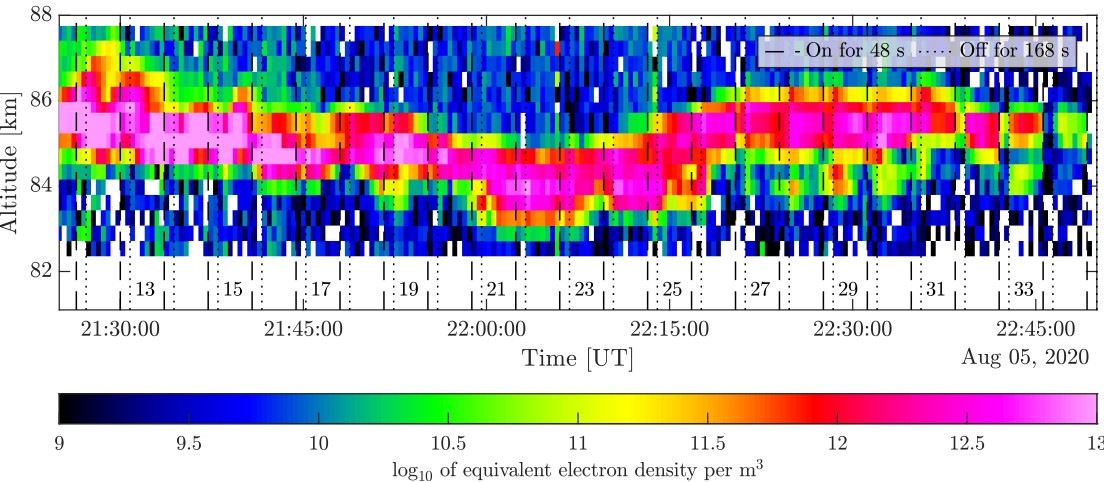

**Figure A7.** Back scattered power as function of altitude and heating intervals observed during the night of August 05/06, 2020 in Area 1.



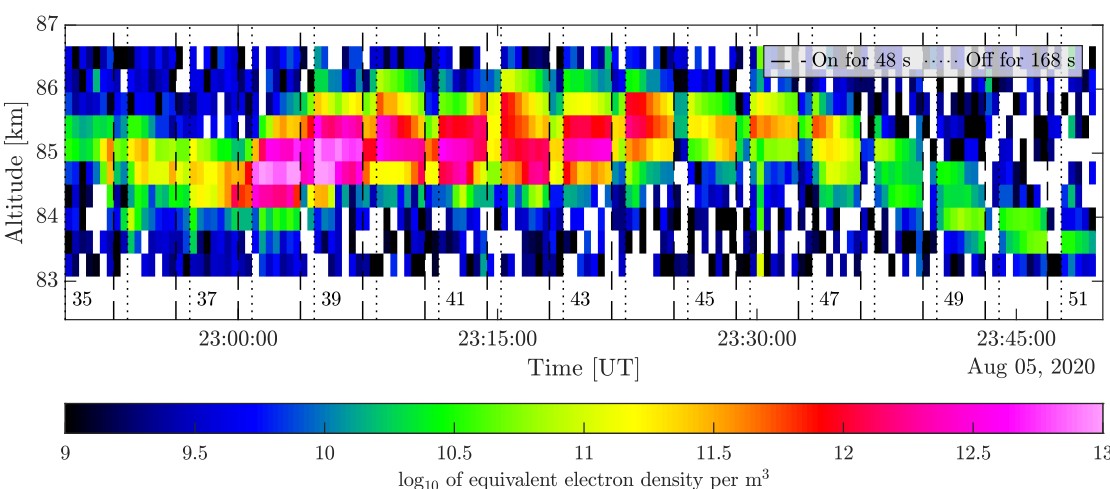

**Figure A8.** Back scattered power as function of altitude and heating intervals observed during the night of August 05/06, 2020 in Area 2.

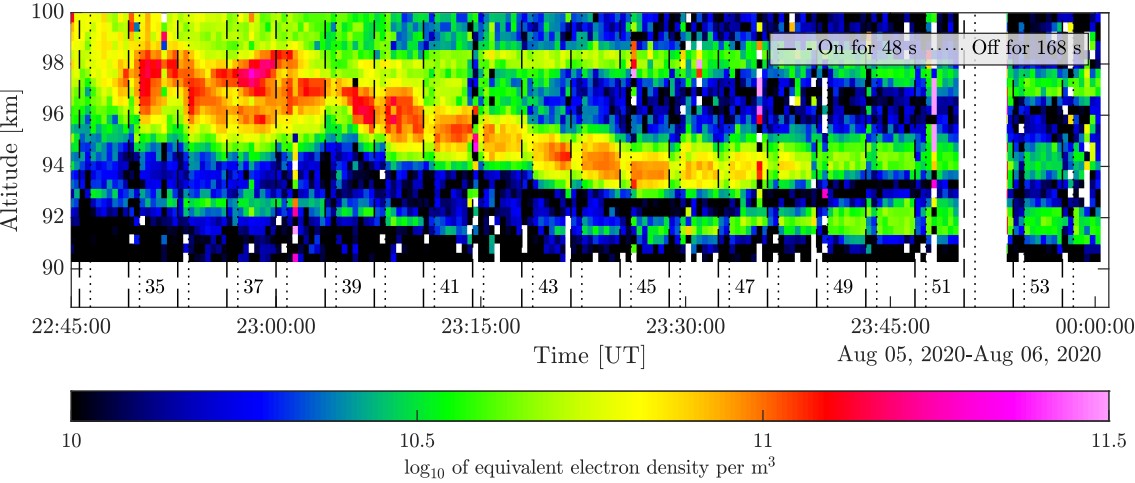

**Figure A9.** Back scattered power as function of altitude and heating intervals observed during the night of August 05/06, 2020 in Area 3. It should be noted that other limits of the colour scale have been chosen for this plot.



## A3    Night of Aug 06/07, 2020

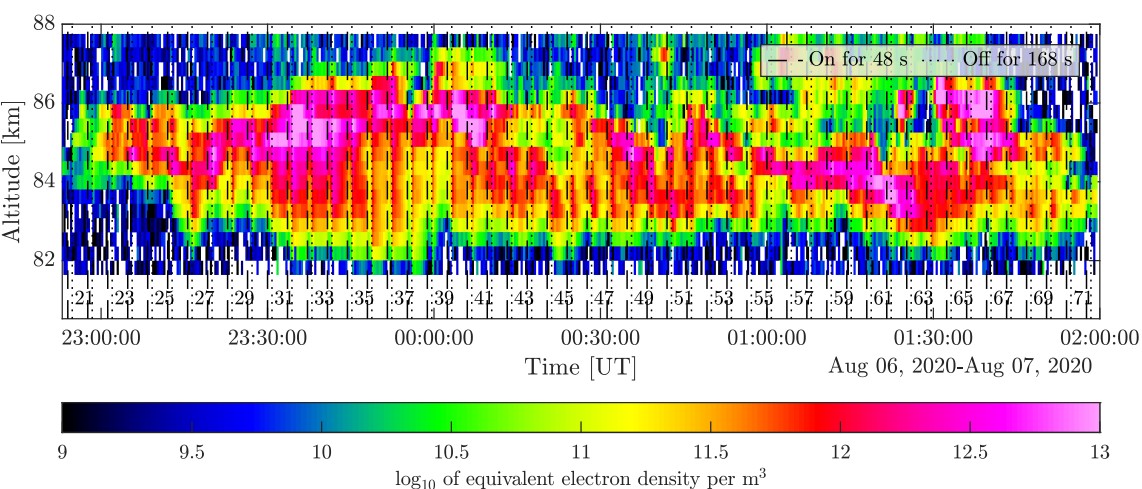

**Figure A10.** Back scattered power as function of altitude and heating intervals observed during the night of August 06/07, 2020 in Area 1.

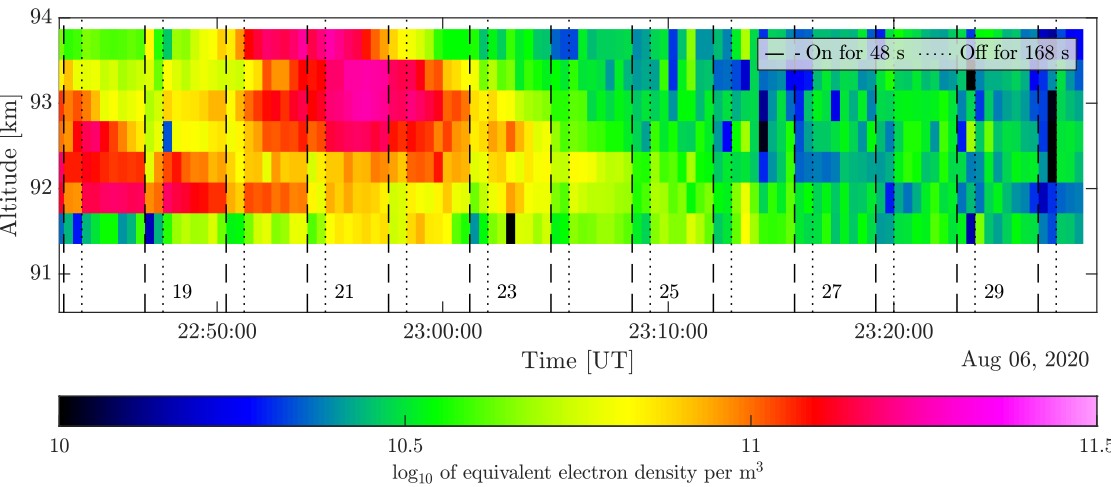

**Figure A11.** Back scattered power as function of altitude and heating intervals observed during the night of August 06/07, 2020 in Area 2. It should be noted that other limits of the colour scale have been chosen for this plot.



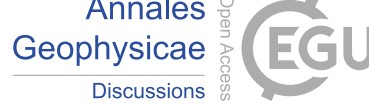

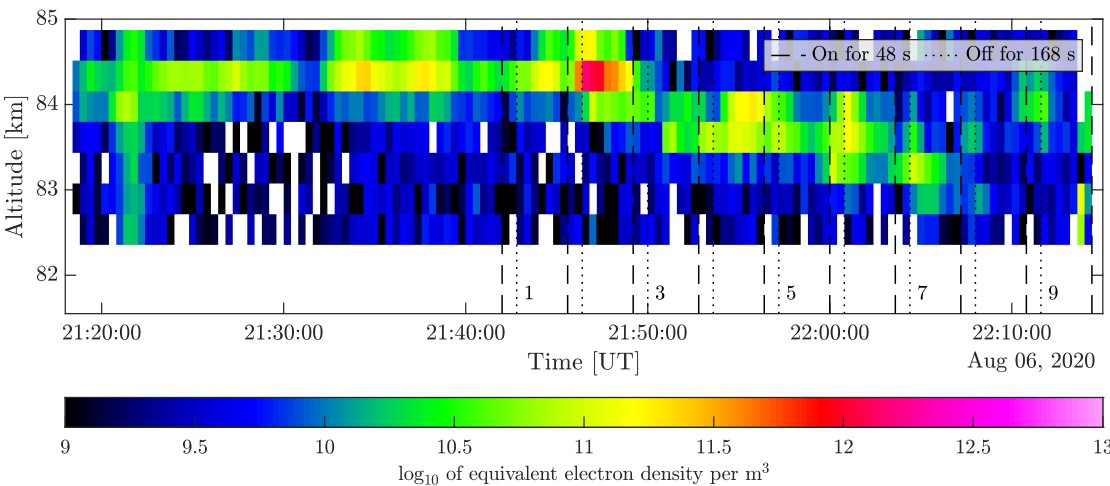

**Figure A12.** Back scattered power as function of altitude and heating intervals observed during the night of August 06/07, 2020 in Area 3.

**Appendix B: Selected detailed PMSE signal**

**B1 Night of Aug 15/16, 2018**

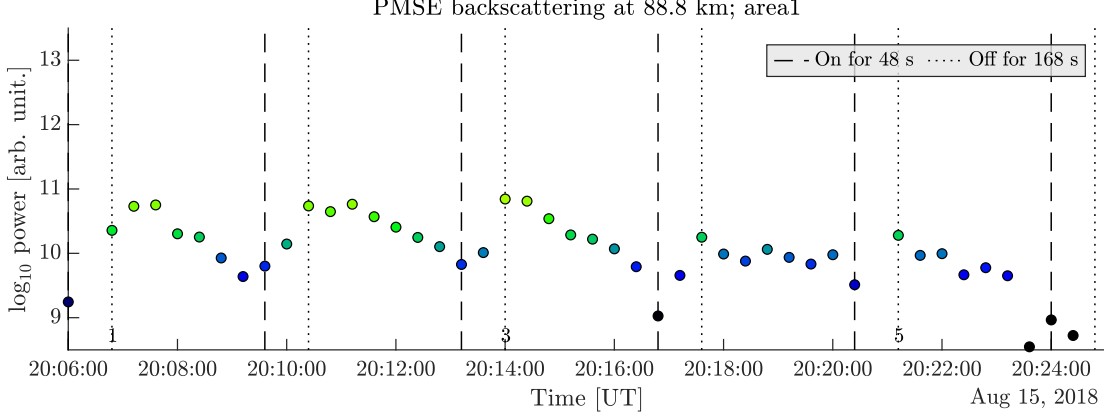

**Figure B1.** Back scattered power at altitude 88.8 km and heating intervals observed during the night of August 15/16, 2018 in area 1. The colour of the dots follow the colour scale of figure A4.



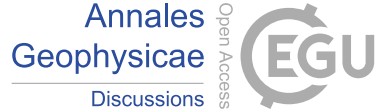

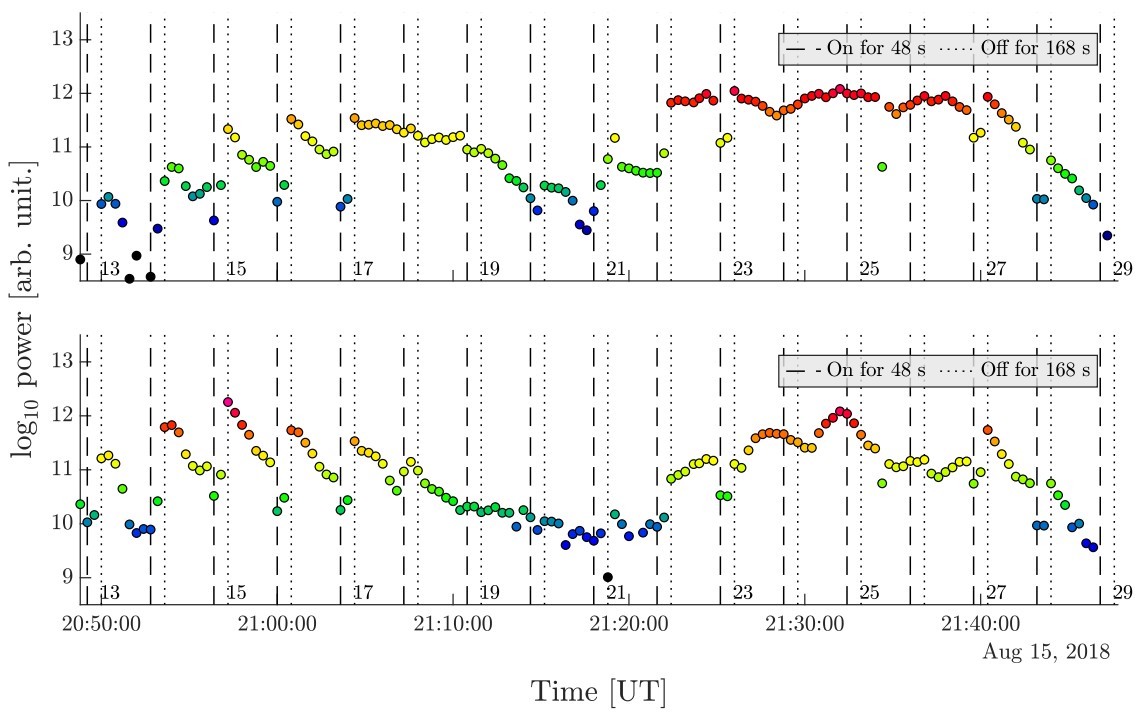

**Figure B2.** Back scattered power at altitude 87.4 and 87.8 km and heating intervals observed during the night of August 15/16, 2018 in area 2. The colour of the dots follow the colour scale of figure A5.



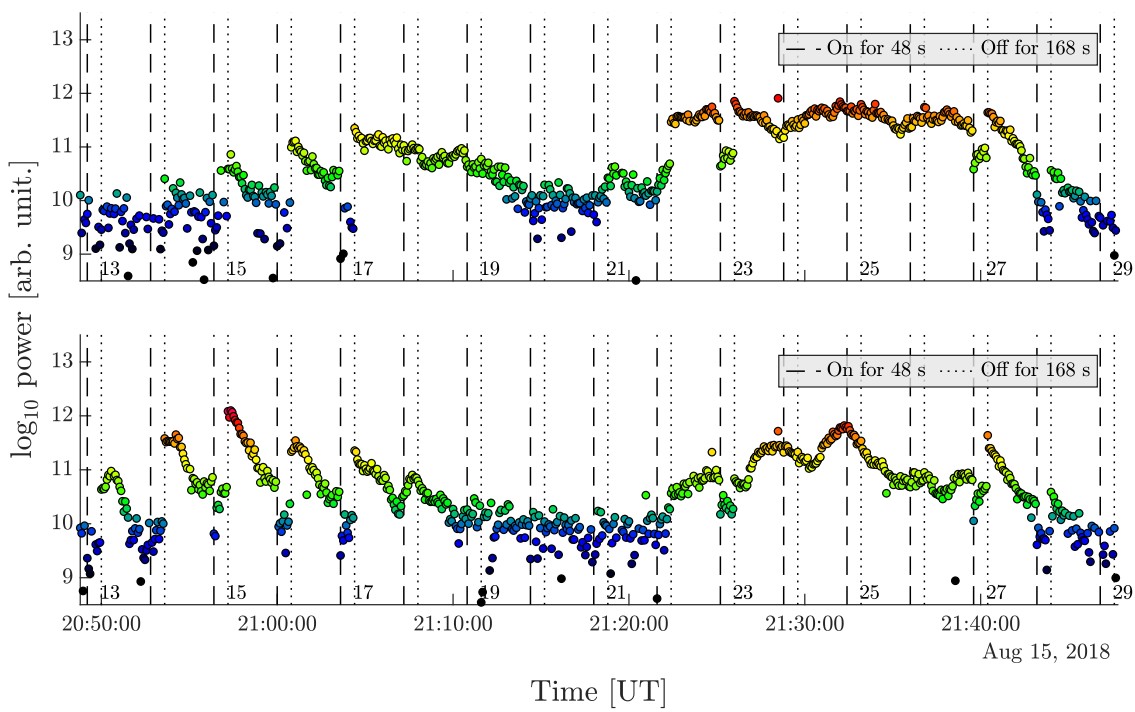

**Figure B3.** Back scattered power at altitude 87.4 and 87.8 km and heating intervals observed during the night of August 15/16, 2018 in area 2. Each 4.8 seconds there is a data point, usually the interval is 24 seconds. The colour of the dots follow the colour scale of figure A5.





## B2    Night of Aug 05/06, 2020

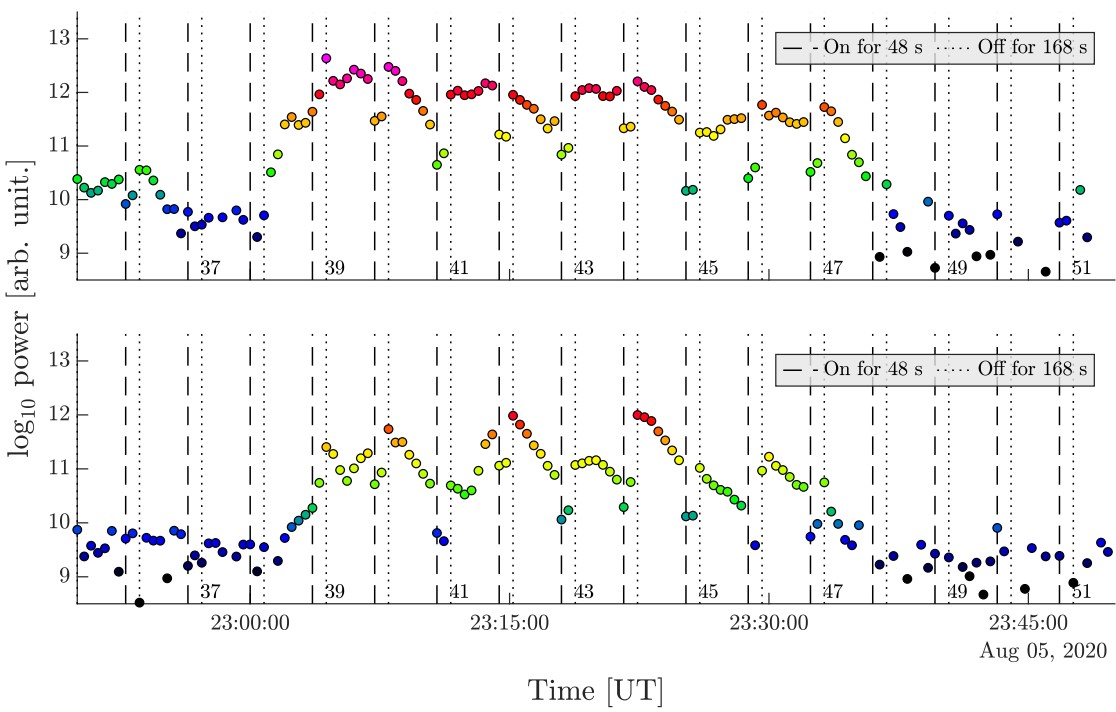

**Figure B4.** Back scattered power at altitude 85.2 and 85.6 km and heating intervals observed during the night of August 05/06, 2020 in area 2. The colour of the dots follow the colour scale of figure A8.





## Appendix C:  Comparison of the power amplitudes

**Figure C1.** Comparison of the power amplitudes observed on the 11 August 2018 in Area 1.







**Figure C2.** Comparison of the power amplitudes observed on the 11 August 2018 in Area 2.





**Figure C3.** Comparison of the power amplitudes observed on the 11 August 2018 in Area 3.



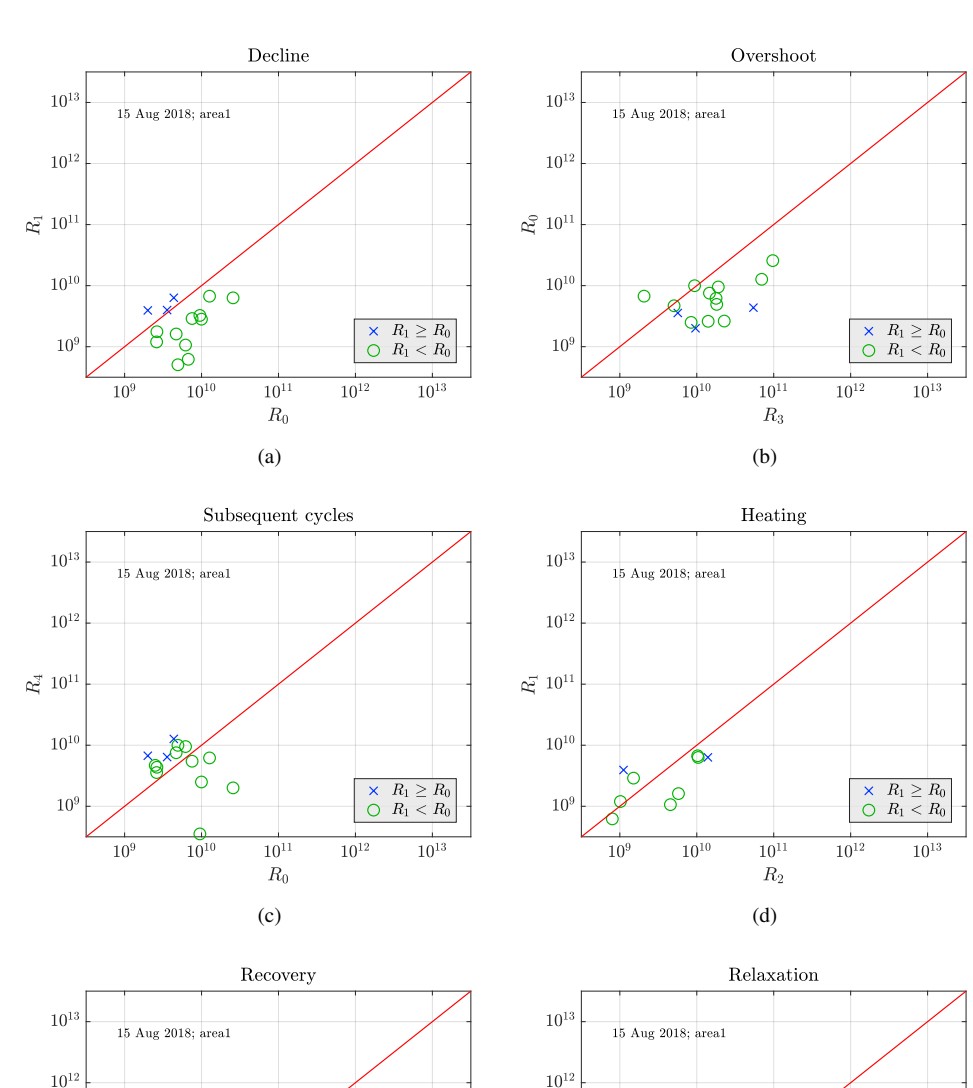

**Figure C4.** Comparison of the power amplitudes observed on the 15 August 2018 in Area 1.





**Figure C5.** Comparison of the power amplitudes observed on the 15 August 2018 in Area 3.





**Figure C6.** Comparison of the power amplitudes observed on the 5 August 2020 in Area 1.







**Figure C7.** Comparison of the power amplitudes observed on the 5 August 2020 in Area 3.





**Figure C8.** Comparison of the power amplitudes observed on the 6 August 2020 in Area 1.





**Figure C9.** Comparison of the power amplitudes observed on the 6 August 2020 in Area 2.







**Figure C10.** Comparison of the power amplitudes observed on the 6 August 2020 in Area 3.





## Appendix D: Histograms of heating cycle ratios with averages

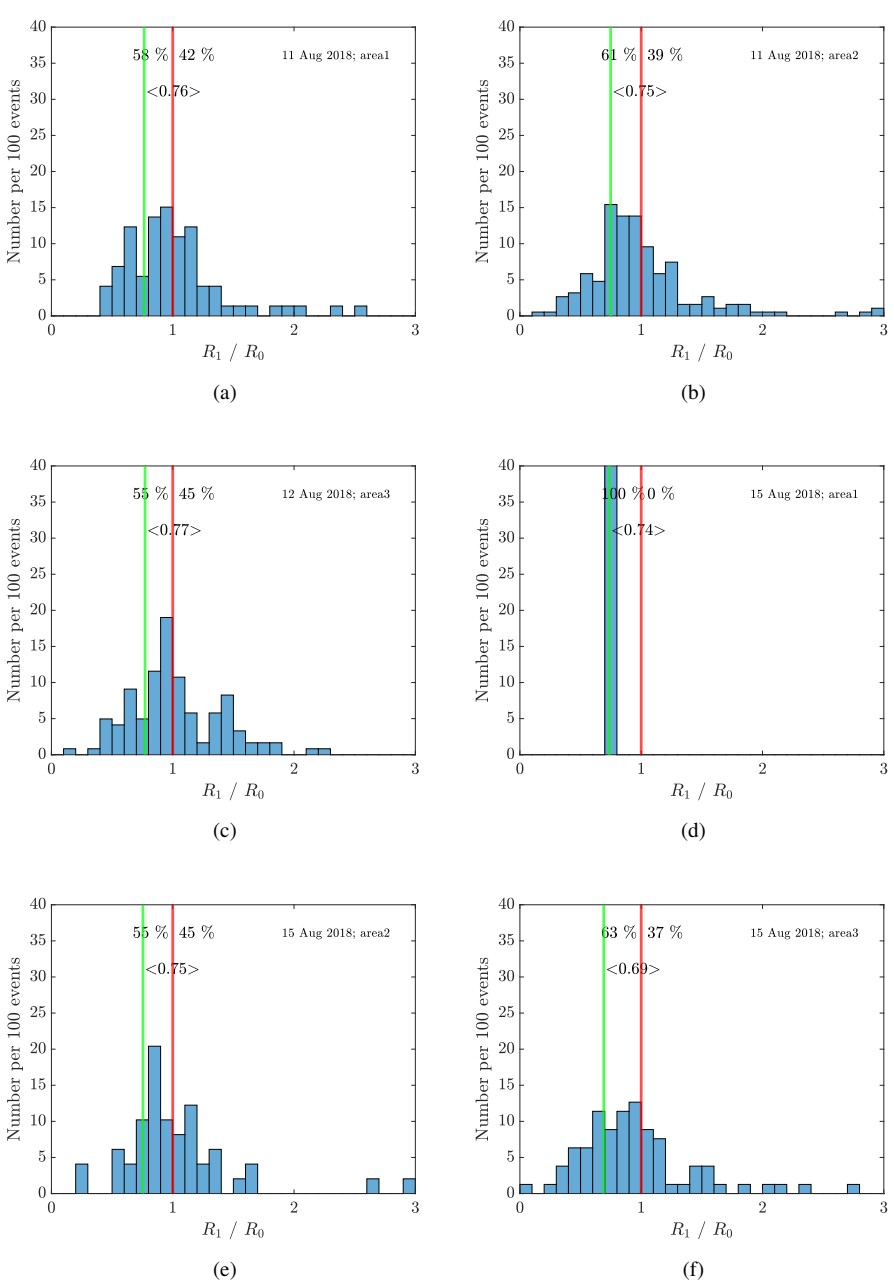

**Figure D1.** Average of the decline for the observed data on the 11 and 15 August 2018. Only overshoot curves with a minimal background amplitude of $R_0 > 10^{10.5}$ are considered. The ratios are chosen in such a way, that, if we observe an overshoot curve like shown in Figure 1, all ratios are smaller than 1. Thus, the histograms are clipped at a maximum ratio of 3. The green line and the corresponding number displays the mean for all ratios smaller than 1.



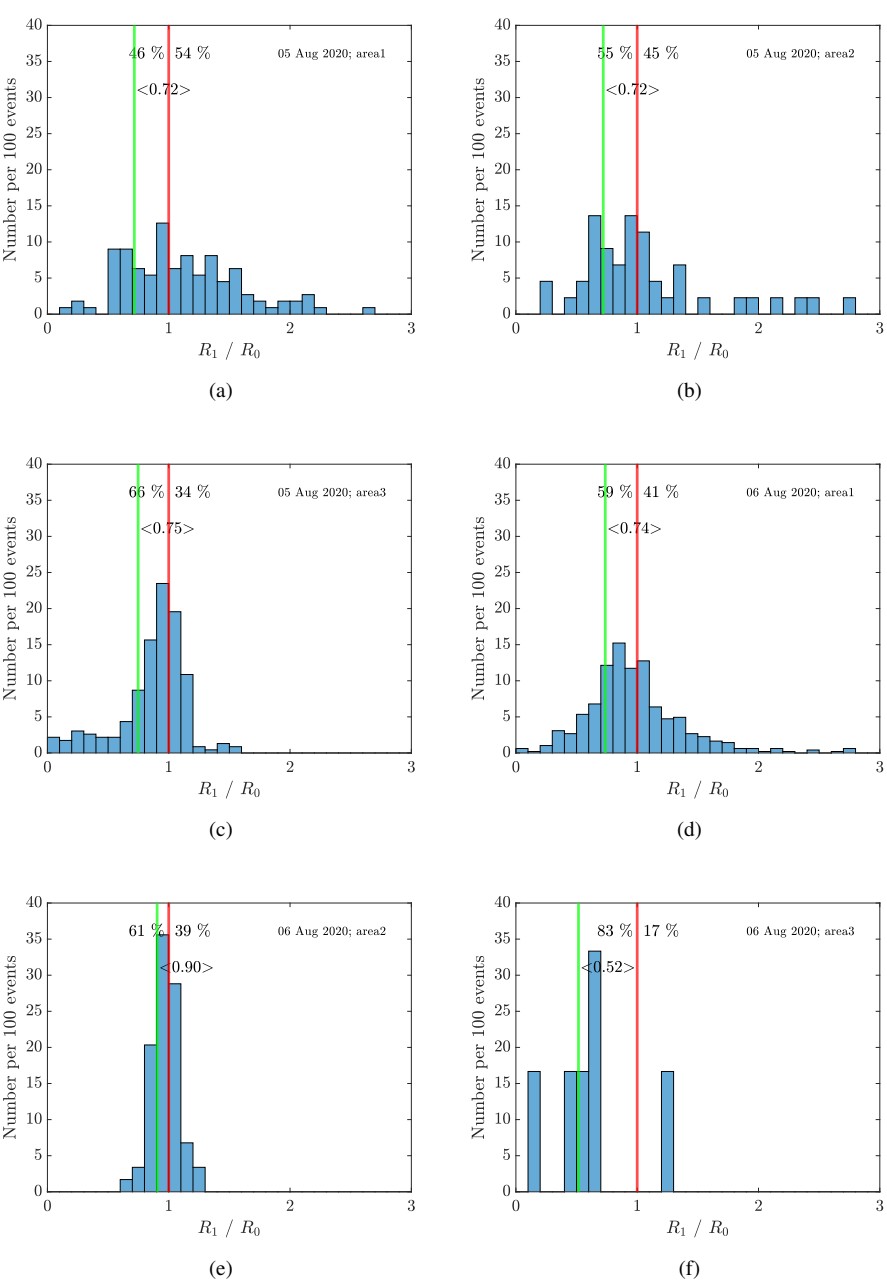

**Figure D2.** Average of the decline for the observed data on the 5 and 6 August 2020. Only overshoot curves with a minimal background amplitude of $R_0 > 10^{10.5}$ are considered. The ratios are chosen in such a way, that, if we observe an overshoot curve like shown in Figure 1, all ratios are smaller than 1. Thus, the histograms are clipped at a maximum ratio of 3. The green line and the corresponding number displays the mean for all ratios smaller than 1.



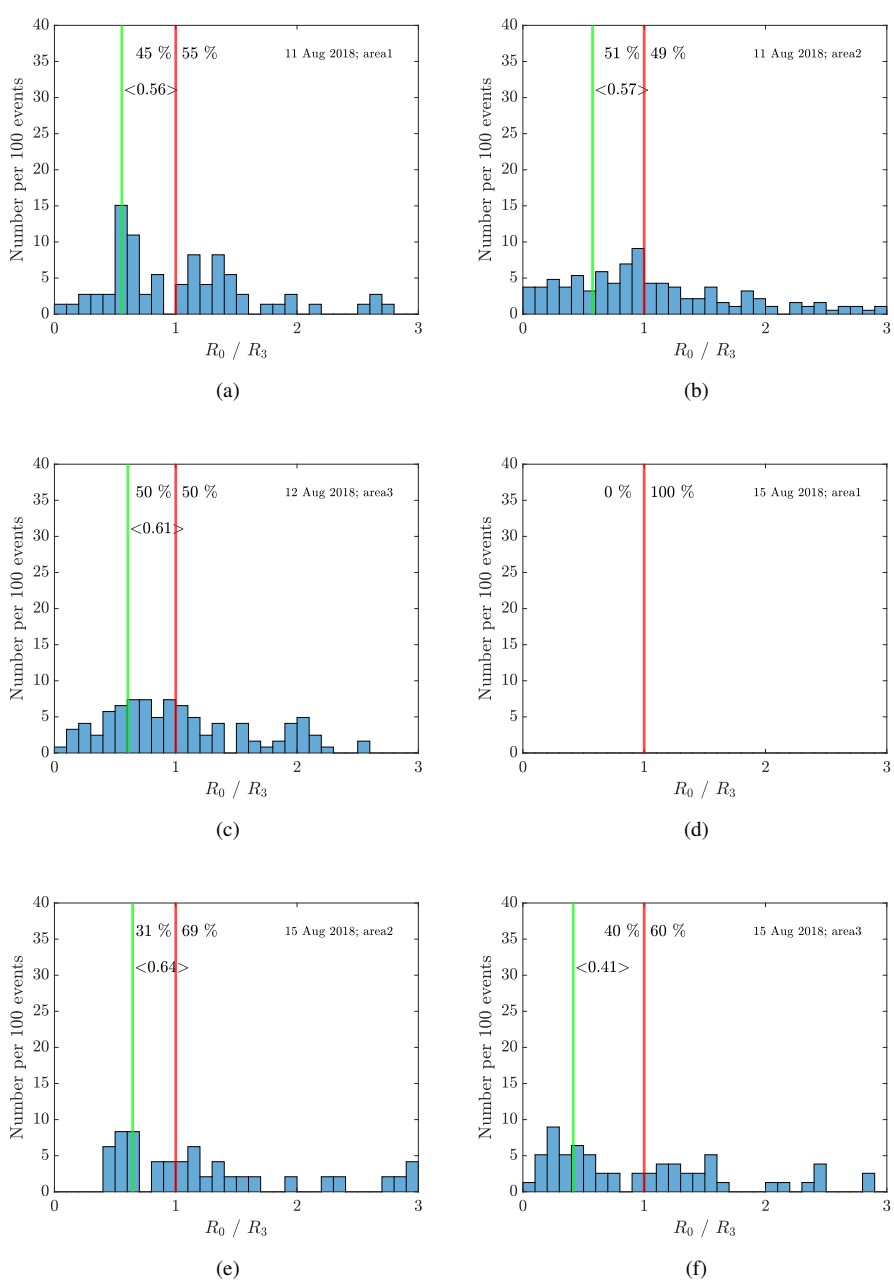

**Figure D3.** Average of the overshoot for the observed data on the 11 and 15 August 2018. Only overshoot curves with a minimal background amplitude of $R_0 > 10^{10.5}$ are considered. The ratios are chosen in such a way, that, if we observe an overshoot curve like shown in Figure 1, all ratios are smaller than 1. Thus, the histograms are clipped at a maximum ratio of 3. The green line and the corresponding number displays the mean for all ratios smaller than 1.



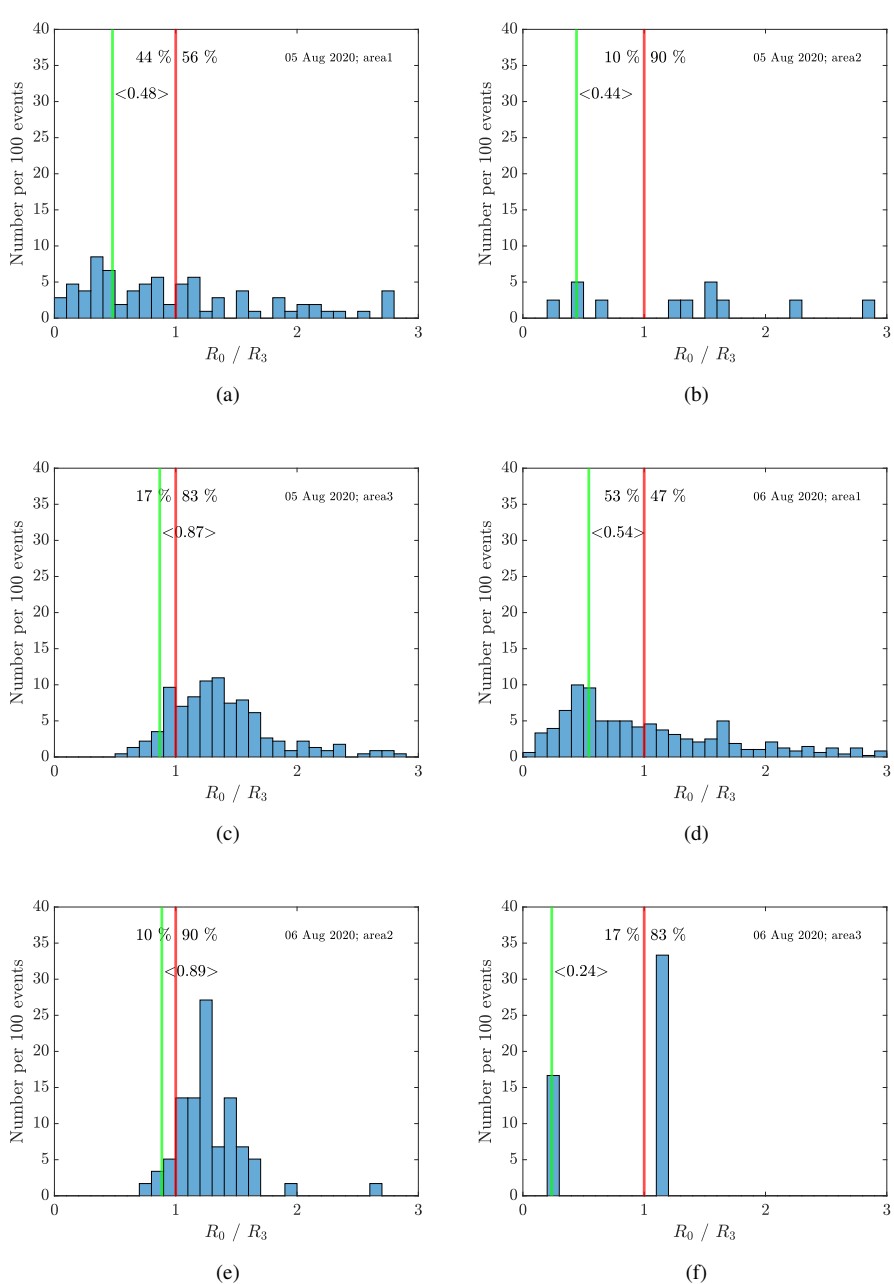

**Figure D4.** Average of the overshoot for the observed data on the 5 and 6 August 2020. Only overshoot curves with a minimal background amplitude of $R_0 > 10^{10.5}$ are considered. The ratios are chosen in such a way, that, if we observe an overshoot curve like shown in Figure 1, all ratios are smaller than 1. Thus, the histograms are clipped at a maximum ratio of 3. The green line and the corresponding number displays the mean for all ratios smaller than 1.



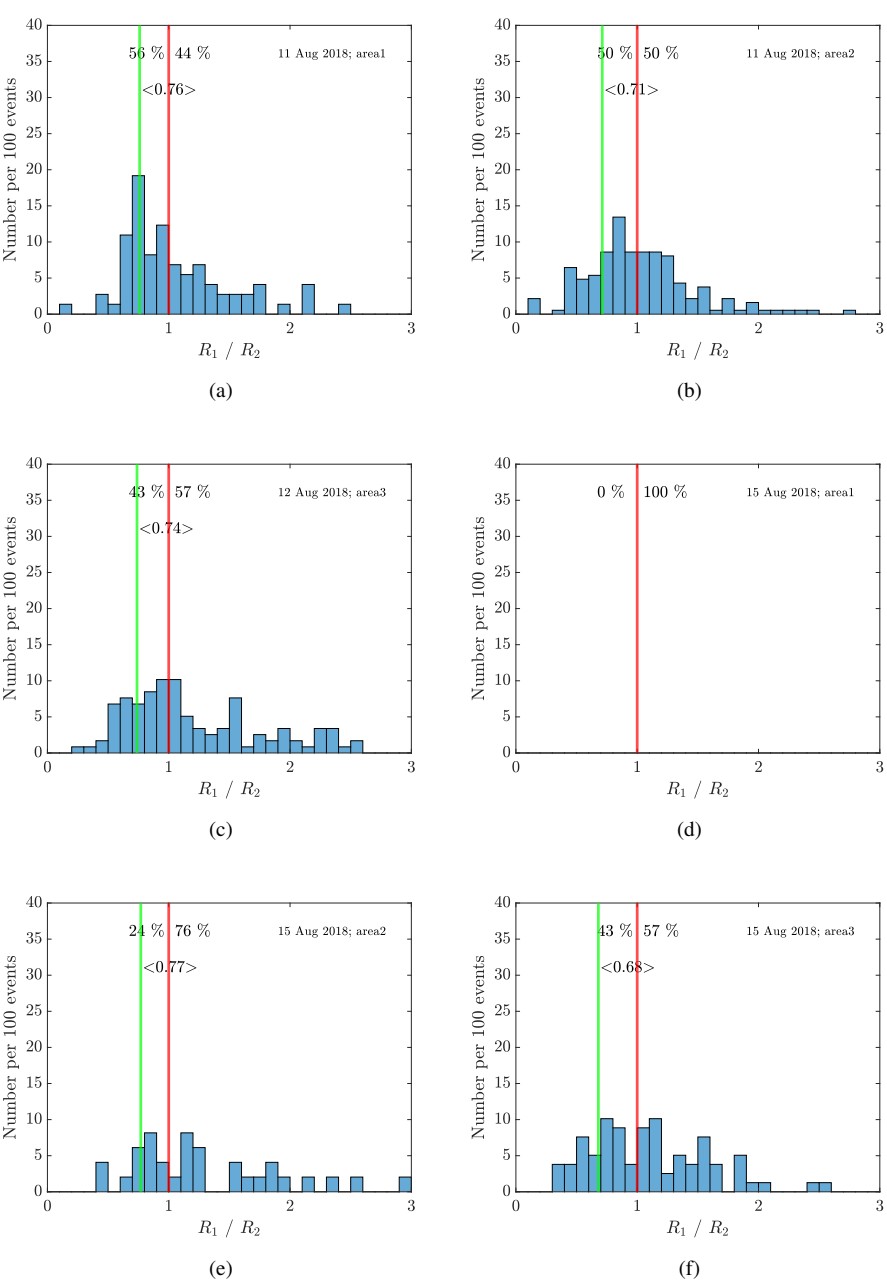

**Figure D5.** Average of the heating for the observed data on the 11 and 15 August 2018. Only overshoot curves with a minimal background amplitude of $R_0 > 10^{10.5}$ are considered. The ratios are chosen in such a way, that, if we observe an overshoot curve like shown in Figure 1, all ratios are smaller than 1. Thus, the histograms are clipped at a maximum ratio of 3. The green line and the corresponding number displays the mean for all ratios smaller than 1.





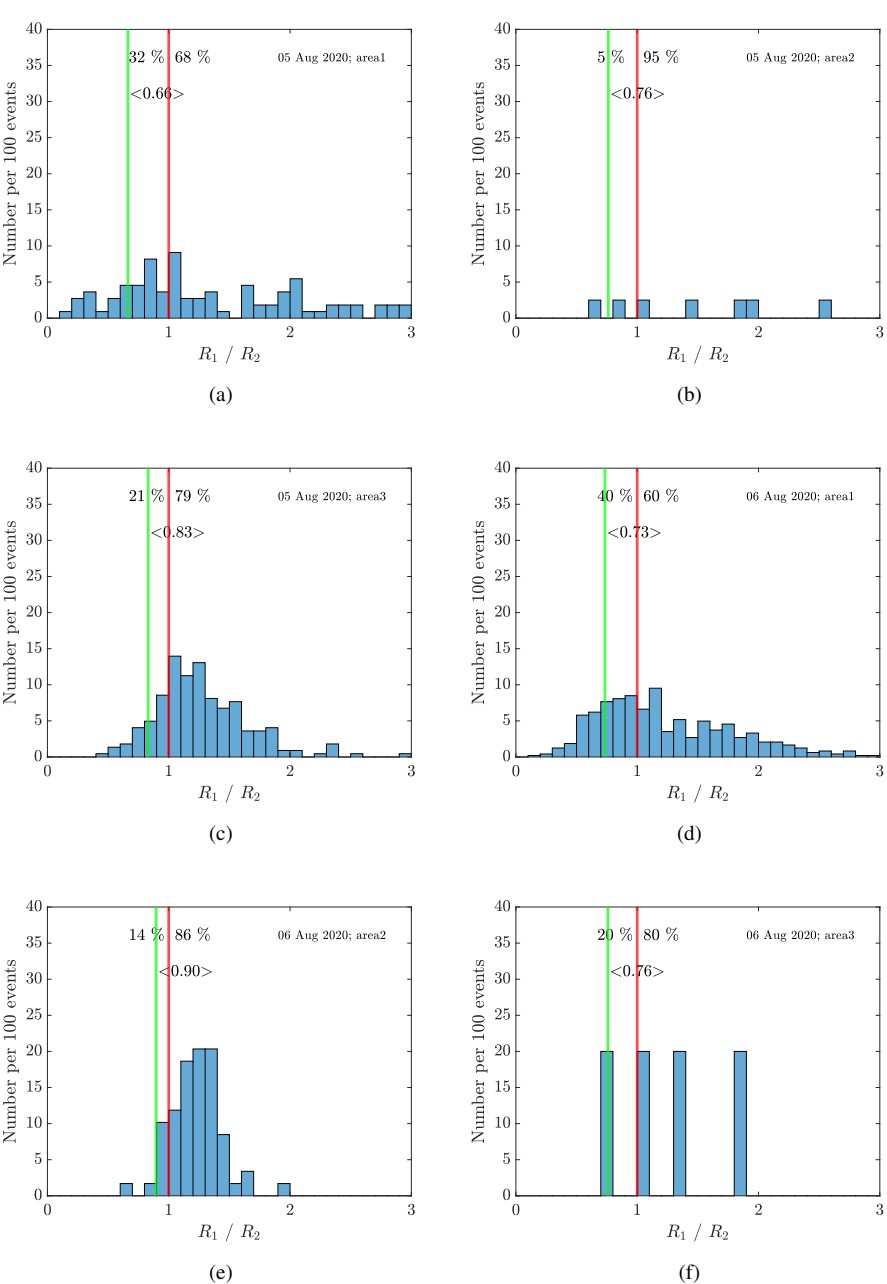

**Figure D6.** Average of the heating for the observed data on the 5 and 6 August 2020. Only overshoot curves with a minimal background amplitude of $R_0 > 10^{10.5}$ are considered. The ratios are chosen in such a way, that, if we observe an overshoot curve like shown in Figure 1, all ratios are smaller than 1. Thus, the histograms are clipped at a maximum ratio of 3. The green line and the corresponding number displays the mean for all ratios smaller than 1.



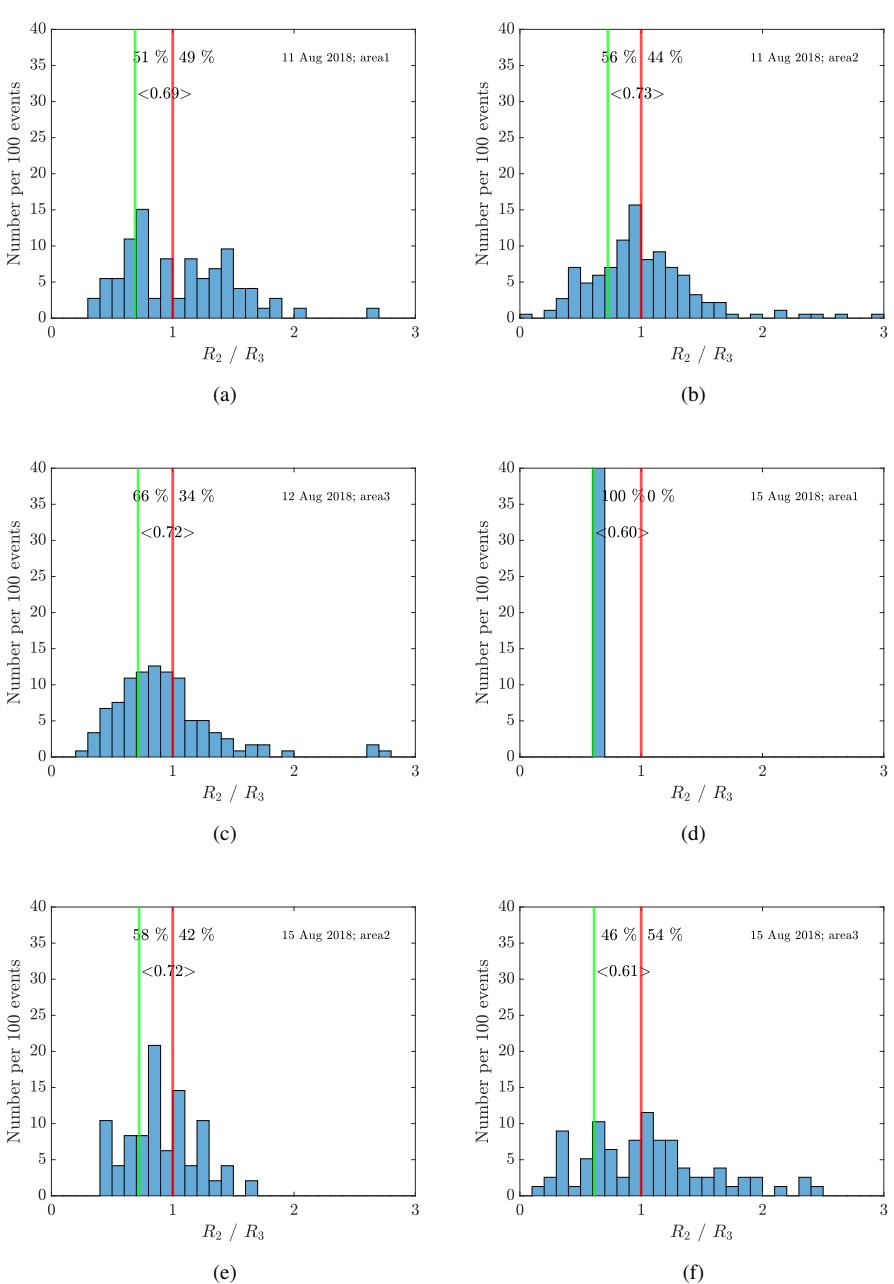

**Figure D7.** Average of the recovery for the observed data on the 11 and 15 August 2018. Only overshoot curves with a minimal background amplitude of $R_0 > 10^{10.5}$ are considered. The ratios are chosen in such a way, that, if we observe an overshoot curve like shown in Figure 1, all ratios are smaller than 1. Thus, the histograms are clipped at a maximum ratio of 3. The green line and the corresponding number displays the mean for all ratios smaller than 1.



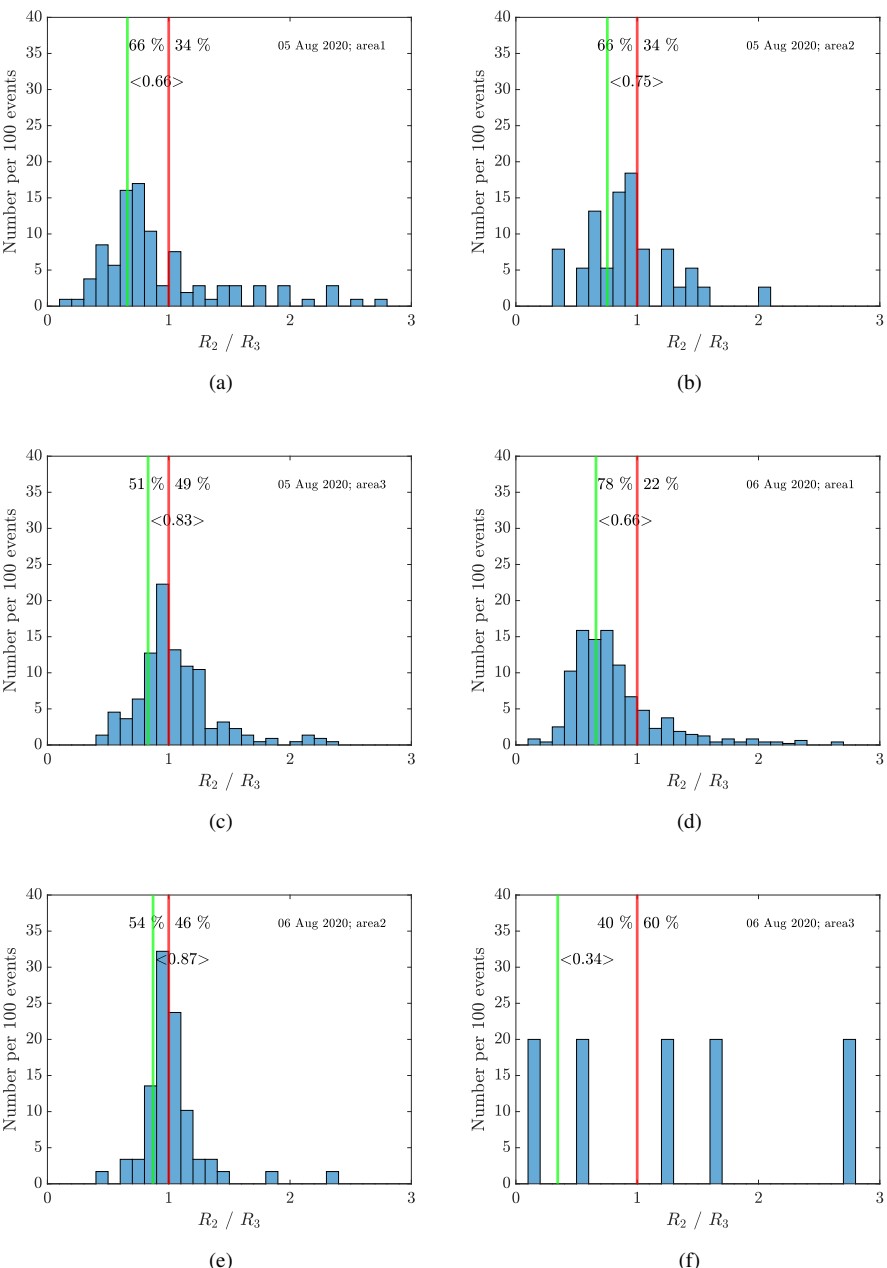

**Figure D8.** Average of the recovery for the observed data on the 5 and 6 August 2020. Only overshoot curves with a minimal background amplitude of $R_0 > 10^{10.5}$ are considered. The ratios are chosen in such a way, that, if we observe an overshoot curve like shown in Figure 1, all ratios are smaller than 1. Thus, the histograms are clipped at a maximum ratio of 3. The green line and the corresponding number displays the mean for all ratios smaller than 1.





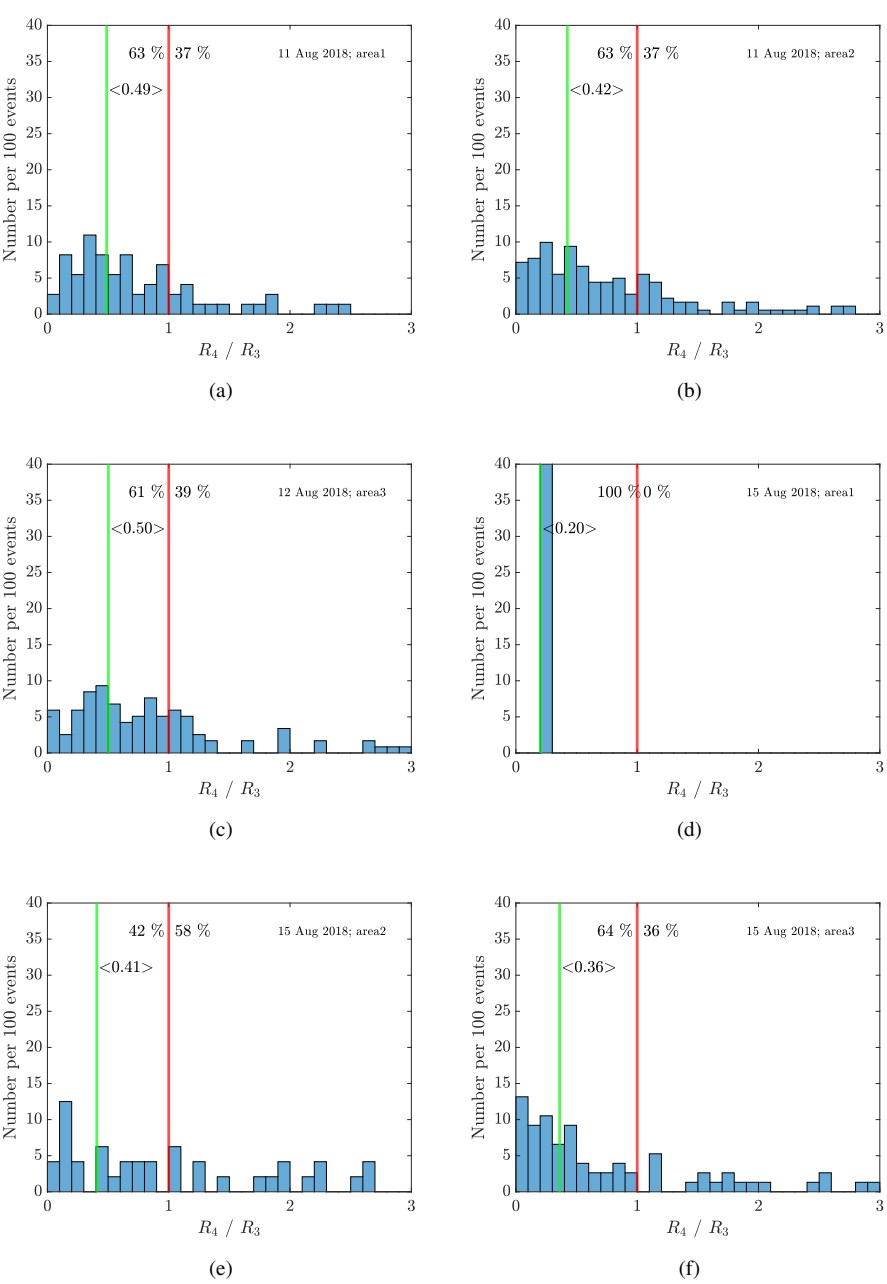

**Figure D9.** Average of the relaxation for the observed data on the 11 and 15 August 2018. Only overshoot curves with a minimal background amplitude of $R_0 > 10^{10.5}$ are considered. The ratios are chosen in such a way, that, if we observe an overshoot curve like shown in Figure 1, all ratios are smaller than 1. Thus, the histograms are clipped at a maximum ratio of 3. The green line and the corresponding number displays the mean for all ratios smaller than 1.





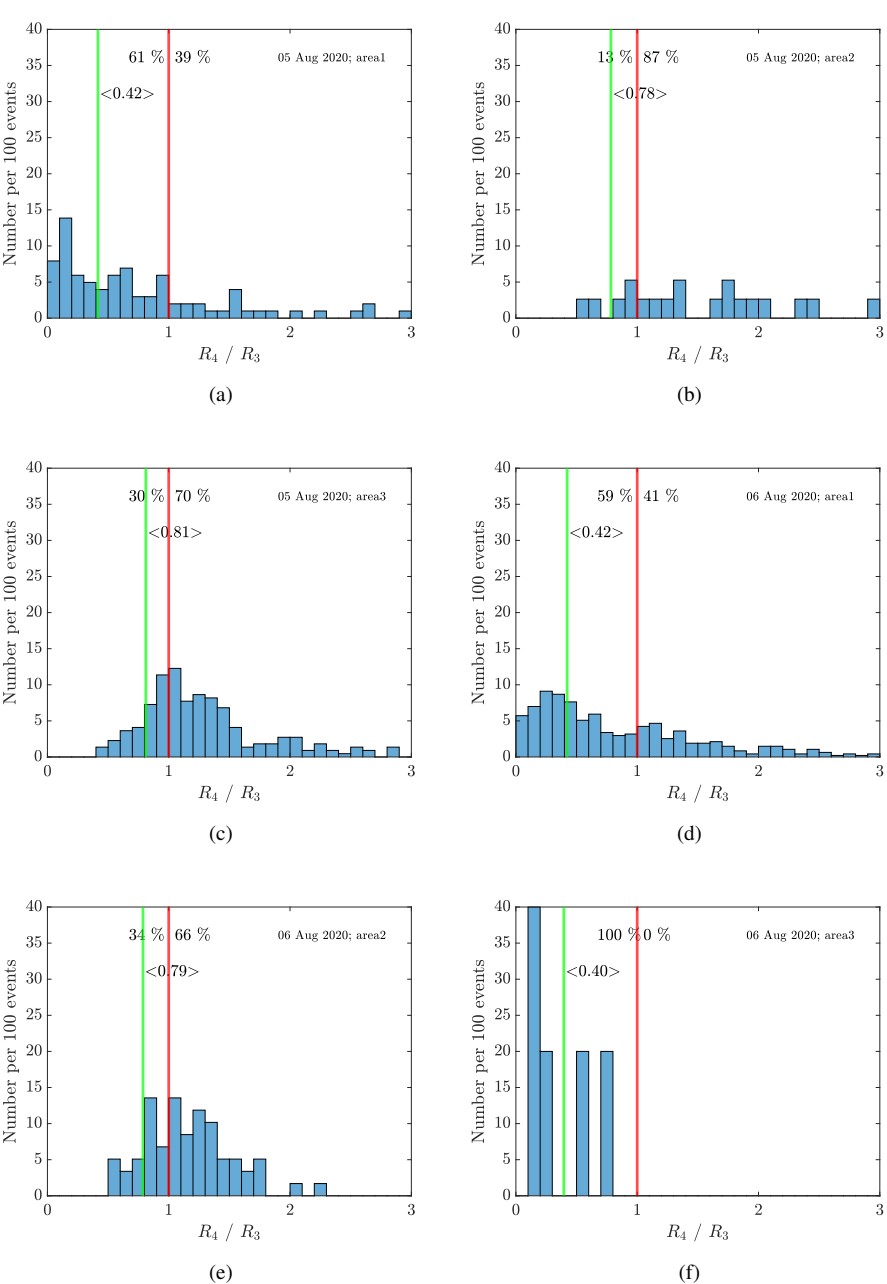

**Figure D10.** Average of the relaxation for the observed data on the 5 and 6 August 2020. Only overshoot curves with a minimal background amplitude of $R_0 > 10^{10.5}$ are considered. The ratios are chosen in such a way, that, if we observe an overshoot curve like shown in Figure 1, all ratios are smaller than 1. Thus, the histograms are clipped at a maximum ratio of 3. The green line and the corresponding number displays the mean for all ratios smaller than 1.