# Peer review of "Modulation of Polar Mesospheric Summer Echoes (PMSE) with HF Heating during low solar illumination"

_Annales Geophysicae, 2022_

## Author Response (AR1)

**Referee 1:**
This paper provides an analysis of PMSE modulation by HF heating. It presents data from several days under varying background ionospheric conditions and analyses the response of the different PMSE layers to heating, including a comparison with simulation to explain the size of the overshoot.

There is some nice, thorough analysis and definitely merit in this work. However, there are also some major issues with the submission, mostly in the way that the paper is presented. The comments I provide below are given in the spirit of trying to improve the manuscript

The main subject:
The first major issue is that it is not at all clear what the main point of the paper is. The title is very generic and gives no clue about what the new/noteworthy and exciting science is. Many papers have been written on HF modulation of PMSE (as the authors show in their introduction), why and what does this one add to the current state of knowledge?

I think that the main takeaway is that this is the first time that analysis has been performed on modulated PMSE under relatively low solar illumination, at late local times and at the end of the traditional PMSE season. The reference to this in the abstract (and how it may affect the observations) is very vague (line 15): "Some individual curves however, show a stronger overshoot than observed in previous studies. A possible explanation for this difference can lie in the dust charging conditions that are different during the night or other conditions might be at play." Essentially you offer a suggestion and then undermine it by saying it might be something else, but without saying what the something else might be.

I highly recommend changing the title to better reflect the main aim/results of the paper (e.g., "Artificial modulation of PMSE during low solar illumination"). I recommend highlighting the unique aspects of the work in the abstract, introduction and discussion. Make it clear why this is different to previous work, or what it does to build on and support it.

We have changed the title to better reflect the topic of the paper.

In connection to this, the authors often digress into discussions of sporadic E (it is even mentioned in the abstract). This paper is about PMSE and I recommend removing the discussion and analysis of sporadic E since it is a distraction to the main topic (see line 173, lines 232 onwards). You highlight the E layers as an interval of interest in figure 3c and discuss it as Area 3 in Observations 3 (line 195). Please remove this, and simply state that "the layers above 90 km after 22:45 UT are not PMSE, they are examples of Sporadic E" perhaps with a reference. I understand the temptation to discuss interesting features, but it only detracts from the main focus; if there is something noteworthy about this interval, I recommend writing a separate article on it.

We have removed all discussion of sporadic E layers.

The style:
The manuscript reads often reads more like a student report rather than a journal article, a clear example of which is the inclusion of a table of contents. The submission includes a sizeable appendix, which is effectively the same size as the main body of the article. I leave it to the editor to determine whether the appendix is appropriate.

We have removed all the figures from the appendix and included them as supplementary material to the paper. They are also now included in the data repository along with the code.

The submission requires significant proof reading for some spelling but mostly for grammar, with attention on the appropriate use of articles ('the', 'a'). I do recognise that English is not the easiest language when it comes to this issue and that it can be far from intuitive, but I note that the paper has experienced authors and at least one who has English as a first language and so I am sure they can be helpful.

Thank you for your comments and suggestions on improvement to the paper. We have tried to make the paper more to the point and with a better focus on the important details. In addition, reviewed the language and grammar.

Additional Comments:

Line 61 you give a range of temperatures (130 – 150 K) and then say, 'and lower'. Why not say "150 K and lower".

Line 61: This is now changed.

Line 69: You state that "The charges and sizes of the dust particles are largely unknown". Is this really true or is it more nuanced than that? Lots of work has been done by Havnes (for example - Havnes, O. and Kassa, M.: On the sizes and observable effects of dust particles in polar mesospheric winter echoes, J. Geophys. Res, 114, D09209, doi:10.1029/2008JD011276, 2009.) on examining dust sizes, others have also done a lot (e.g., Rapp, Scales, etc)

Line 69: The sizes and possibly also the charge states are possibly different between summer and winter echoes but yes you are right, this sentence makes it sound like there is no previous work done when that is not the case. We have now changed this section to better reflect previous work.

Line 71: 'shown' instead of 'suggested'

Line 71: Changed.

Line 130 onwards: you describe the observations taking place under dusk and night conditions. This gives the initial impression of some of the observations taking place in darkness, but are they really? You go onto discuss the reduction in solar illumination of an order of magnitude compared to midday in mid-Summer, but I am not sure if a reader would have a feel for what this really means. To put it in context, how does the range of illumination vary through the year (lowest to highest) – where on the range do these observations sit? Did you consider including a sentence on the solar zenith angle (just the average if it doesn't vary much) at the times of the observations? This is a measure that I suspect the readers may be more familiar with and are more likely to intuitively understand the difference in degrees between mid-summer and these observations.

Given that the level of illumination is potentially the major selling point of the paper, it would be wise to make sure the reader understood how unique this might be in terms of past observations.

Line 130: Yes thank you for highlighting this point. The solar zenith angles are all in the range 88-97 degrees with the main part of the observations larger than 90 degrees. This would mean that the sun is below the horizon but only by a few degrees. This essentially means that the observations are still in sunlight but greatly reduced due to the change in optical depth. We have included a range of the solar zenith angles in question as well as a small figure depicting the estimated photon flux for the Lyman alpha line for 21.06 at noon and for 15.08 at 22 and 24 hours in order to better show the reader the amount of reduced sunlight.

Line 138: Is it possible to give an estimate of the change in the photo-emission current? Or indicate that is something that you will consider in the analysis. Saying it 'should translate…' without providing more detail seems very vague at this stage.

Line 138: We now show the reduction in the photon flux from 21.06 to 15.08 (at different times),  which is an order of magnitude and more. The photo-emission current is proportional to the photon flux and thus reduces by an order of magnitude as well.

Line 139 – 144: this section seems to describe observations that were mostly irrelevant. Except for the later comment about lack of NLC limiting likelihood of dust size I would have recommended removing this. This could be shortened to: "weather conditions were poor for measuring NLC, though cameras were operating at Kiruna and Nikkaluokta (~200km south of Tromso) to allow triangulation. During 15/16 August, faint NLC were visible from Kiruna, close to the horizon, but westward of the EISCAT site (closer to Andoya)"

I do have a question about this: how confident are you that the lack of NLC above Tromso was not due to low clouds in the line of site of the camera? This is quite important since your dismissal of larger particle sizes later on relies on this.

Line 139-144: We have shortened the text in question and changed the conclusion. We cannot conclude that there were no NLC present for the observations with bad weather. For the 15-16 of august 2018 observation however, there were no NLC visible above the EISCAT site.

Figure 2 and lines 145 onwards: these measurements have been selected as they are closest to the radar measurements at the time of the PMSE observations. This is unacceptably vague, how far away were they? 1 degree of latitude, 2, 10? It is important to give the right context for the reader to judge.

Figure 2 and lines 145+: For Fig.2a, the horizontal distance between these profiles and Tromsø is 490 km. For Fig. 2b, the horizontal distance is 293 km. The geographical coordinates are in the figures but we forgot to specify this. We have included this information in the text for the reader, both the distance as well as the coordinates. We also specify that these conditions might not apply for the EISCAT site.

Lines 153-160: I thought it might be useful to include an example of some of the grammatical issues that occur throughout the paper, along with an indication of some alternate wording and removal of over-precise measurements.

"The radar observations were made in the zenith direction with the EISCAT VHF (224 MHz) antennas radar near Tromsø (69.59°N, 19.23°E). The VHF and Heating parameters are given in Table 1. Details on the radar experiments were provided by the EISCAT documentation

(Tjulin, 2022*). EISCAT Heating (Rietveld et al., 1993, 2016) was operated with a vertical beam at 5.423 MHz with a nominal 80 kW per transmitter, which corresponds to Effective Radiated Power (ERP) between 500 and 580 MW. and X-mode polarization was used with HF pumping in a sequence of 48 s on and 168 s off. The beam width of the Heating facility (7°) is much larger than that of the VHF radar (~4°) and given that the vertical winds and velocity fluctuations of the PMSE are only a few m/s and horizontal winds possibly upto a few 10s m/s (Strelnikova and Rapp, 2011), the radar is always measuring PMSE influenced by the heating."
*include the documentation as a reference, but make sure you pick the relevant edition for the year of the experiments.

Lines 153-160: Thank you for your example, we have reviewed the text as a whole and made corrections to the language. We have also added a reference to the EISCAT documentation.

Line 167: remove "we consider the back-scattered power in units…" since you have effectively said this on line 163.

Line 167: It is now removed.

Line 185 onwards: I am curious about the early part of this observation, it is very hard to see because of the scales, but even the weak precipitation from 2000 – 2110 UT seem to shows signs of modulation at higher altitudes. I know that precipitation can show distinct natural modulation that leads to regular Ne patterns, but I wonder if this is a problem with the analysis not fully taking into account that the system power is fluctuating as the heater is turned on and off.

Line 185 onwards: GUISDAP uses the measured radar transmitter power for the analysis which we believe is reliable and correct. The vertical lines of modulation seen on 15 Aug 2018, between about 90 km to the top at 110km are heater-induced and are almost certainly electron heating of the whole 90-110 km altitude region as the heater wave propagates through this part of the ionosphere, which is only lightly ionized.

I would also be wary about suggesting the PMSE in (1) 'seems that it was trigged by the heating', beyond the coincidence of timing is there a reason for this speculation?

Our suggestion here was that if there would not have been heating there most likely would not have been PMSE visible in area 1 and that the overshoot caused by the heating make the faint PMSE visible. We have removed this, since it is only a speculation.

Power modulation might also be visible in Observation 3 around 21 UT. Is it possible that this might also be responsible for the modulation of the sporadic E later in that interval, there are hints the striations continue to higher altitudes? I recommend talking to your coauthors Reitveld and Haggstrom to double check whether this might be the case.

These lines at 21 UT are most likely broad-band interference caused by arcing in the heating transmission line/antenna system.

Line 220 onwards: if these figures are important enough to discuss in the main text, why are they not included in the main body of the paper?

Line 220+: The figures are quite many and all show histograms with averages. We have compiled the main results in the referenced table instead and changed the text to refer to the table and then for the interested reader the figures can be seen.

Line 223: why was this value selected? You discuss the threshold is perhaps too high, in which case why did you not try the analysis with a different threshold and publish that?

Line 223: We use this threshold since other authors have used it to indicate the presence of PMSE and thus we can better compare our results to theirs. In addition, by lowering the threshold we would introduce many instances of data dominated by noise.

Line 262 and elsewhere: when you refer to particular data that you have presented, you should refer to that figure (i.e. fig3c)

Line 262 and elsewhere: Yes we made sure to reference all figures.

Lines 388 onwards: I found the description a bit confusing. When you say that the 'simulations assume an initial plasma temp...' do you mean that you set that as the initial temperature based on the IRI value for that time? Or did you mean that the simulations are hard wired with that temperature which thankfully happened to match what IRI would expect? I assume not the latter but that is sort of how it reads. Also, how confident are you that the IRI temperature is itself reasonable. Did you look at the EISCAT temperatures for heater-off times around the PMSE region to get an estimate for comparison? How did you determine that 3nm dust particles were the best fit – did you judge it by eye, or did you fit the model to the data by altering the dust size until some measure of best fit was achieved?

Lines 388 onwards: The simulations are set at 150 K and the heating temperature calculated as a multiple of that initial temperature. By using 150 K the simulations can be compared to previous publications that have simulated the overshoot since many use 150 K as an initial temperature. This value was also quite close to both the IRI model values as well as the satellite temperature measurements from figure 2. EISCAT temperature estimates are from the IRI model, and not measured.

Figure 16. Why is the X-axis scale different between the data and the simulation?

Figure 16: This is just to show that the simulation goes back to equilibrium when you let it run longer. This is also what we expect from the simulations. For interval A the simulation suggests that this takes longer than what the observations suggest.

Line 472 onwards: is there a way to explore how much of an effect absorption of the heater beam might play? The way this is written you seem to be suggesting that you have an answer for the large overshoot in interval A, but that other effects may be important and perhaps even fully explain the large overshoot, thus negating your initial statement. It is really good to consider other factors and suggest alternative explanations, however it is better if you can provide an indication of how confident you are in your explanation and why. As it stands it sort of reads as if you are saying: "analysis suggests that X can explain the overshoot effect, but it could also be explained by Y instead". Stating uncertainties and explaining the limits of analysis are of course very good, but at the5 same time you can make it clearer how confident you are in the analysis.

"we know that Y potentially contributes to the effect at an unquantified level; however our analysis shows that N% of the effect is explained by X"

Line 472+: Yes you are right, we have reviewed our statements here and hopefully made it more clear. Our conclusion is that the high overshoot is caused by a combination of both the absorption of the heater and an increased electron temperature in addition to a shorter diffusion timescale.

I was surprised that there is no reference to this, who also found varying overshoot for different conditions (albeit with different radar frequencies as well):

Havnes, O., Pinedo, H., La Hoz, C., Senior, A., Hartquist, T. W., Rietveld, M. T., and Kosch, M. J.: A comparison of overshoot modelling with observations of polar mesospheric summer echoes at radar frequencies of 56 and 224 MHz, Ann. Geophys., 33, 737–747, https://doi.org/10.5194/angeo-33-737-2015, 2015.

Thank you for this suggestion to the Havnes et al (2015) paper; it is a very interesting paper and should have been included. We have fixed this now.

Summary. There is some good analysis and interesting data in this submission. Some work is required to bring the level of presentation up to standard and in doing so highlight the important aspects of the work. The title needs rethinking, and the key results need to be better highlighted in the abstract, introduction and discussion to make them stand out more. I think its important to check that the guisdap analysis has properly taken the system porwer changes into account in each instance.

I am confident that the authors can address these issues and produce a good paper.

We have made several changes to the language and to the aforementioned sections of the paper. Thank you for your time and comments.

**Referee 2:**

The authors display various aspects of observed PMSE under articial HF heating and stretching over ~4-6 hours each of several nights and a thorough analysis of the data. A simulation complements this analysis. Generally I also think that the main points do not become immediately obvious to the reader, owing to the large amount of presented material and the sometimes lengthy and too pondering language. Appendices show material that the authors consider as details and perhaps less important, which I think is a good division. Nevertheless, some "streamlining" of the opus could improve it.

We the authors thank you Stephan C. Buchert for your comments and help in making our manuscript better. We have made several corrections to the paper.

Specifically, Figure 3c and line 198: "Area 1" starts at 21:30 UT, but the PMSE seems be present already at the start of the measurements at 20:30 UT, partially with precipitation. What is the reason that this period was not included in "Area 1" or added as separate "Area"? The period would be perhaps be long enough to make a small difference for some of the statistical analysis.

Figure 3c and line 198: Unfortunately due to operational problems with the heater this part of the observation could not be included in the analysis. Most likely due to broad-band interference caused by arcing in the heating transmission line/antenna system.

Line 454: "... λirreg is the wavelength of the irregularity and is given by $2*\lambda R$ (the radar wavelength) ..." According to the Bragg condition (https://en.wikipedia.org/wiki/Bragg%27s_law) the irregularity wavelength would be half of radar wavelength , i. e. "given by $\lambda R/2$", just 67 cm for the EISCAT VHF. Is this just a textual mistake, or are the columns "τdff at 0 sec" and "τdff at 48 sec" of Table 3, I think, too large by a factor 4? According to my rough calculation the latter is the case.

Line 454: Yes you are right, it is λirreg = $\lambda R/2$. An unfortunate mistake that has caused the calculated diffusion timescales to be much too large.

Which values of Te/Ti and Zd*nd0/ne0 in equation (1) were used to produce the numbers is Table 3?

Te/Ti: Were the temperatures obtained from the GUISDAP analysis? In the standard configuration GUISDAP would always give values Te=Ti≈150 K. However, the HF heating increases Te dramatically, which is the main cause of the suppression of PMSE at "heater on" and the overshoot following "heater off". This increase of Te probably cannot be estimated from the radar data. Chen and Scales (2005) had assumed that Te/Te0 (=Te/Ti) would reach 10. But the values in Table 3, ratio of columns "τdff at 0 sec" and "τdff at 48 sec", seem to be inconsistent with such large heating. A plot of the for the simulation assumed Te/Te0 over time might be helpful.

Similarly, which model of dust charging was simulated/assumed in the simulations? The numbers in Table 3 suggest that Zd*nd0/ne0 did not change much between 0 and 48 sec, but the overshoot would depend strongly on the amount of dust charging?

These minor issues perhaps don't affect the main conclusions of the paper, but should of course be checked and explained. Perhaps assuming a larger Te/Te0 and stronger dust charging could explain the large observed overshoots?

No the temperature ratio is taken from the simulation and not the radar data. Values for Te/Ti at 0 seconds is assumed to be 1 before the heater is turned on, for values at 48 seconds we use the ratios given in figure 16 (simulation on right). Te/Ti = 8 for interval A and Te/Ti = 5 for intervals B and C. Values for Zd come from figure 17 (left) taken at 0 seconds and 48 seconds and nd0/ne is taken from given values in figure 16(right)

For the estimation of tdff in table 3 only Zd is assumed to change from 0 sec to 48 sec not the ratio nd0/ne0 where we have used the value for Zd from figure 17. The ratio for nd0/ne0 we have used the values from figure 16(right) and assumed that it is the same for both 0 sec and 48 sec.

We have decided to remove table 3 from the manuscript, as its main purpose was to examine the neutral density effect of the diffusion timescale however the diffusion time scale should be calculated using numerical results and not estimated in a few instances as we have done here. We will rather focus on the figures that show the ratio of the two timescales connected to the simulation and only mention involvement of neutral density.

I tried to verify some of these details by looking at the code, but the link given on line 529 does not work for me.

We apologize for the link not working we will make sure to fix this. Thank you for your discussion and comments.

**Main changes to the paper related to referees comments:**

1. Reviewed the language and grammar of the paper and made several changes where appropriate.
2. Included a new figure in chapter 3.1 Overall observation conditions. This figure shows the decrease in photon flux for 15 August at 22 h and 14 h compared to 21.06 at noon. We have also included the range of solar zenith angles relevant for the observations.
3. We have removed the discussion of sporadic E layers.
4. In chapter 4 we have moved Overall observational discussion to 4.3 instead of 4.1.
5. We have removed the estimates and related equations of the diffusion timescale in table 3/chapter 4.4.
6. We have included 3 more references:

- Havnes, O., Pinedo, H., La Hoz, C., Senior, A., Hartquist, T. W., Rietveld, M. T., and Kosch, M. J.: A comparison of overshoot modelling
  with observations of polar mesospheric summer echoes at radar frequencies of 56 and 224 MHz, Annales Geophysicae, 33, 737–747,
  https://doi.org/10.5194/angeo-33-737-2015,
  https://angeo.copernicus.org/articles/33/737/2015/, 2015
- Pinedo, H., La Hoz, C., Havnes, O., and Rietveld, M.: Electron–ion temperature ratio estimations in the summer polar
  mesosphere when subject to HF radio wave heating, Journal of Atmospheric and Solar-Terrestrial Physics, 118, 106–112,
  https://doi.org/https://doi.org/10.1016/j.jastp.2013.12.016,
  https://www.sciencedirect.com/science/article/pii/S1364682613003349, recent progress from networked studies based around MST radar, 2014
- Tjulin, A.: EISCAT experiments, EISCAT Scientific Association,(March), 2017.

7. We have removed many of the figures from the appendix since it became extremely long. The figures are now added in the repository along with any relevant coed to reproduce the figures as well as a supplement to the paper.